# Mesa-Extrapolation: A Weave Position Encoding Method for Enhanced Extrapolation in LLMs

**Xin Ma[1], Yang Liu[2,3], Jingjing Liu[2], Xiaoxu Ma[1]**
[1]Digital Research Institute, Enn Group, Beijing, China
[2]Institute for AI Industry Research, Tsinghua University, Beijing, China
[3]Shanghai Artificial Intelligence Laboratory, China
{xin.ma0206, xiaoxuma}@gmail.com, {liuy03, jjliu}@air.tsinghua.edu.cn

## Abstract

Large language models (LLMs), although having revolutionized many fields, still suffer from the challenging extrapolation problem, where the inference ability of LLMs sharply declines beyond their max training lengths. In this work, we conduct a theoretical analysis to better understand why *No Position Encoding* (NoPE) fails outside its effective range, as well as examining the power of *Position Encoding* (PE) in this context. Our findings reveal that with meticulous weave position, PE can indeed be extended beyond effective range. Our theorems establish that LLMs equipped with *weave PE* can achieve improved extrapolation performance without additional cost. Furthermore, we introduce a novel weave PE method, *Mesa-Extrapolation*, which utilizes a chunk-based triangular attention matrix and applies *Stair PE* to manage the final chunk. This method not only retains competitive performance but also offers substantial benefits such as significantly reduced memory demand and faster inference speed. Extensive experiments validate the effectiveness of Mesa-Extrapolation, demonstrating its potential as a scalable solution to enhancing LLMs' applicative reach. Our code is available at `https://github.com/soacker/Mesa-Extrapolation`.

## 1 Introduction

Large Language Models (LLMs) with their powerful in-context learning capabilities Brown et al. (2020) offer versatile solutions to a wide-range of intelligent applications. However, one pressing challenge, the *extrapolation problem* Delétang et al. (2022) Zhang et al. (2022), dictates that the inference ability of LLMs sharply declines beyond their max training lengths, imposing a serious limitation on applications with long inputs. An naive solution is to extend the length of training samples. However, the inherent quadratic complexity of calculations presents practical challenges, demanding more resources, longer training time and higher cost.

Positional encoding (PE) has become a pivotal component of Transformer architecture, to compensate for the overlooking of position information by the attention mechanism. In the realm of extrapolation capability, PE is considered as a key factor influencing the extrapolating ability of LLMs. A few PE approaches, such as the popular RoPE Su et al. (2023) and ALiBi Press et al. (2021), claim to offer improved extrapolation capabilities and have gained widespread usage in industrial applications. Meanwhile, a counter-narrative has emerged. Some works demonstrate that Transformer can achieve better extrapolation capabilities by removing position encoding (NoPE) Kazemnejad et al. (2023), and contend that the mask already plays a significant role in capturing position information.

---

[*]Corresponding author

38th Conference on Neural Information Processing Systems (NeurIPS 2024).

Inspired by Kazemnejad et al. (2023), we conduct a thorough investigation of the extrapolation problem by designing a specific Transformer model for this purpose and presenting detailed theoretical analysis. To the best of our knowledge, this is the first theoretical endeavor to understand the inner workings of extrapolation. We first elucidate the cause of NoPE's failure when input exceeds the effective window length in Theorem 3.1, and analyse PE's failure in 3.2. Building upon Theorem 3.2, we prove in Theorem 3.3 that through meticulous weave position, (i.e., *weave PE*), it is feasible to achieve extrapolation beyond the effective window length.

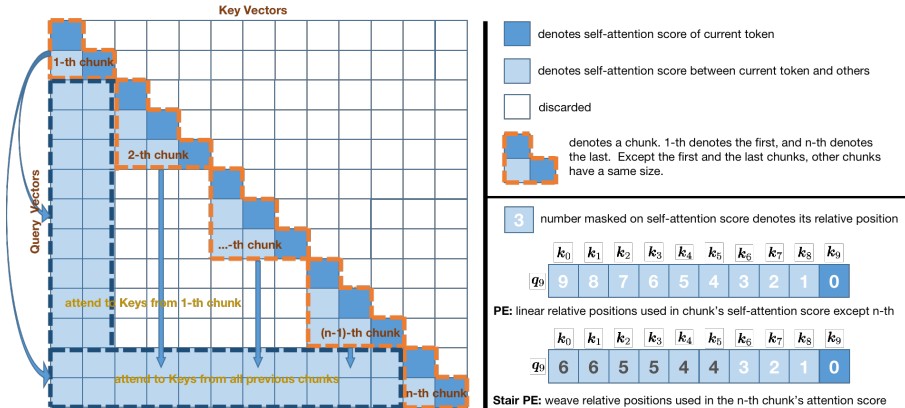

Figure 1: Chunk-based triangular attention matrix, PE and Stair PE. The left figure shows the Chunk-based triangular attention matrix (before SoftMax operation) of Mesa-Extrapolation when an exemplar sequence of length 13 is fed into a LLM. The right figure shows an example of PE and Stair PE. The Stair PE is used to weave the relative position in Mesa-Extrapolation.

We further introduce a novel weave-PE-based method, *Mesa-Extrapolation*, which demonstrates substantial extrapolation prowess without the need for additional training. Specifically, we use chunk-based triangular attention matrix, aiming at memory-friendly resource consumption (Figure 1 left), and we employ Stair PE to handle the last chunk (Figure 1 right below), effectively making Mesa-Extrapolation a completely free plug-in for LLMs.

Our contributions are summarized as follows:

1. **Theoretical Analysis** We provide a comprehensive theoretical analysis on the challenges encountered by NoPE beyond its effective window length, and investigate the effect of PE in this context. Our theorems prove that through meticulous weave position, PE can be effectively extrapolated beyond the effective window length. This theoretical foundation establishes a clear understanding of the extrapolation problem and potential solutions for LLMs utilizing PE.

2. **Introduction of Mesa-Extrapolation** We introduce a novel extrapolation approach called *"Mesa-Extrapolation."* Based on triangular attention matrix, this method strategically organizes input tokens into chunks, with the final chunk employing a weave PE method (e.g., Stair PE) to integrate all token states. Mesa-Extrapolation provides a practical and effective solution to the extrapolation problem.

3. **Empirical Validation** Comprehensive experiments demonstrate that our approach achieves competitive performance, offering the benefits of extremely low memory usage and the fastest inference speed compared to existing methods. Importantly, it is an easy plug-in and can enhance extrapolation performance of LLMs without any additional resource consumption.

## 2 Background

Since the self-attention mechanism itself does not contain position information, PE components have become an integral part of the Transformer architecture. Cmmon choices for PE are either *absolute*, where each absolute position (e.g., $1, 2, 3, ...$) is directly represented, or *relative*, where the distance between tokens is used as positional information. Absolute Position Embedding (APE) embeds

each absolute position $i$ into position vector $\boldsymbol{p}_i$ and adds word embeddings to their corresponding $\boldsymbol{p}_i$, before feeding them to the LLMs Vaswani et al. (2017). However, the extrapolation ability is limited because it cannot generalize to unseen positions. RoPE Su et al. (2023) rotates the query and key vectors with an angle proportional to their absolute positions, so the attention dot production only depends on relative distance between tokens, providing a relative positional encoding. T5's Relative Bias first maps the relative distance $(i - j)$ between tokens at positions $i$ and $j$ to a scalar bias value $b = f(i - j)$, where $f$ is a lookup table. ALiBi is similar to T5's Relative Bias but instead subtracts a scalar bias from the attention score Press et al. (2021)(refer to Appendix D.2 for more details). Recent works such as ReRoPE and Leaky-ReRoPE Su (2023b) achieve effective extrapolation without fine-tuning by meticulously weaving relative positions of RoPE. **We refer to this class of methods that achieve extrapolation by weaving the relative positions of PE without fine-tuning as** *Weave PE*.

Kazemnejad et al. (2023) indicates that the decoder-only transformer with position encoding removed (NoPE) demonstrates stronger extrapolation capabilities. Furthermore, it theoretically shows that a specific transformer model can get relative and absolute positional information, even in the absence of PE. Haviv et al. (2022) also demonstrates NoPE achieves comparative performance to standard Transformer models. **These new studies pose a key challenge regarding the choice of whether using PE or not in Transformer architecture** (refer to Appendix A for more related works).

## 3   Model Extrapolation: NoPE vs. Weave PE

### 3.1   Problem Definition

We mainly consider relative PE methods and formally define their self-attention dot product as a function $f_{\mathrm{PE}}$, which takes the query $\boldsymbol{q}_t$ located on position $t$, the key $\boldsymbol{k}_i$ located on position $i$, and their relative positions $t - i$ as input parameters, as follows:

$$\langle \boldsymbol{q}_t, \boldsymbol{k}_i \rangle := f_{\mathrm{PE}}(\boldsymbol{q}_t, \boldsymbol{k}_i, t - i), \tag{1}$$

where $f_{\mathrm{PE}}$ denotes a relative PE method such as RoPE or ALiBi.

For ALiBi,

$$\langle \boldsymbol{q}_t, \boldsymbol{k}_i \rangle := f_{\mathrm{ALiBi}}(\boldsymbol{q}_t, \boldsymbol{k}_i, t - i) = \boldsymbol{q}_t^T \boldsymbol{k}_i - (t - i) \cdot C^{m+1},$$

where $m$ is head index and $C = 2^{-2^{-\log_2(\#\mathrm{heads}+3)}}$.

For NoPE,

$$\langle \boldsymbol{q}_t, \boldsymbol{k}_i \rangle := \boldsymbol{q}_t^T \boldsymbol{k}_i.$$

Based on Equ.1 we formally define **weave PE** as follows:

$$\langle \boldsymbol{q}_t, \boldsymbol{k}_i \rangle := f_{\mathrm{weavePE}}(\boldsymbol{q}_t, \boldsymbol{k}_i, t - i) = f_{\mathrm{PE}}(\boldsymbol{q}_t, \boldsymbol{k}_i, \mathcal{W}(t - i)), \tag{2}$$

where $\mathcal{W}$ is a weave function which takes the relative position $t - i$ as input parameter.

For example, ReRoPE Su (2023b) can be considered as an example of weave PE, with its $\mathcal{W}$ function defined as follows:

$$\mathcal{W}(t - i) := \left\{ \begin{array}{ll} t - i & , \quad t - i \leq N \\ N & , \quad t - i > N \end{array} \right.$$

where $N$ is a constant. ReRoPE's dot-product attention is:

$$\langle \boldsymbol{q}_t, \boldsymbol{k}_i \rangle := f_{\mathrm{ReRoPE}}(\boldsymbol{q}_t, \boldsymbol{k}_i, t - i) = f_{\mathrm{RoPE}}(\boldsymbol{q}_t, \boldsymbol{k}_i, \mathcal{W}(t - i)) = \boldsymbol{q}_t^T R^{\mathcal{W}(t-i)\theta} \boldsymbol{k}_i,$$

where $R$ is a rotation matrix that rotates $\mathcal{W}(t - i)\theta$ radians. This is based on RoPE's dot-product attention:

$$\langle \boldsymbol{q}_t, \boldsymbol{k}_i \rangle := f_{\mathrm{RoPE}}(\boldsymbol{q}_t, \boldsymbol{k}_i, t - i)) = \boldsymbol{q}_t^T R^{(t-i)\theta} \boldsymbol{k}_i.$$

### 3.2   Motivation

Chen et al. (2023) explores the evolution of hidden state values and reveals a noticable phenomenon: as the position increases, the hidden state values will explode. This finding appears consistent with observed failures in extrapolation. Through probe experiments (refer to Appendix F), we investigate

the alterations in hidden state values across various positions and layers. Results show a significant shift in the hidden state's value range upon surpassing the effective window. Interestingly, when employing extrapolation techniques, hidden state values exhibit noticeable suppression. This indicates that the effective range of hidden state values likely lies within a specific threshold. When the position exceeds the effective window length, the hidden state values surpass this threshold, resulting in extrapolation failures.

Previous work Kazemnejad et al. (2023) utilizes a constructive approach. By constructing the Transformer's weights, it enables the first and second layers to independently generate position information. Drawing inspiration from this, we endeavor to construct a Transformer model capable of mirroring this observation.

### 3.3 Theoretical Analysis

In a multi-layer neural network, each layer's outputs, a.k.a hidden state values $o$, become the inputs for the subsequent layer. To maintain stable network behavior, these values must remain within a reasonable range. **We define this observable boundary as the threshold $\mathcal{H}$.** This threshold can be either an upper bound or a lower bound. For our analysis, we focus on the lower bound of this threshold. A **successful extrapolation** occurs when a large model consistently generates accurate next tokens for a long input sequence. Conversely, a **failed extrapolation** happens when the model produces incorrect or nonsensical next tokens. Based on these definitions, we make the following assumptions:

> **Assumptions.** In LLM, there is a lower bound as threshold $\mathcal{H}$ for the hidden state value $o$ in specific dimension and specific layer. Let $M$ be the max window length for LLM. Predefine query $\boldsymbol{W}_Q$, key $\boldsymbol{W}_K$, value $\boldsymbol{W}_V$ and output $\boldsymbol{W}_O$ matrices, and feed-forward sub-layer $\boldsymbol{W}_1$, $\boldsymbol{W}_2$ matrices. When $o > \mathcal{H}$, LLM extrapolates successfully. Once $o < \mathcal{H}$, LLM extrapolation fails.

These assumptions indicate that by observing whether the hidden state value $o$ in this dimension exceed the threshold $\mathcal{H}$, we can predict whether the large model's extrapolation has failed. Building upon these assumptions, theoretical results for NoPE exceeding the effective window length are as follows:

> **Theorem 3.1** (NoPE Extrapolation). *Let $x = [<bos>, x_1, \ldots, x_T]$ be an input sequence of length $T + 1$ to the model. Then, there exists $\boldsymbol{W}_Q$, $\boldsymbol{W}_K$, $\boldsymbol{W}_V$, $\boldsymbol{W}_O$, $\boldsymbol{W}_1$, and $\boldsymbol{W}_2$ matrices, such that when $T < M$, $o_T > \mathcal{H}$; and when $T > M$, $o_T < \mathcal{H}$.*

Full proof is given in Appendix E.1. This theorem reveals the internal mechanism of NoPE extrapolation as the input length changes. The theoretical results for PE are as follows:

> **Theorem 3.2** (PE Extrapolation). *Let $x = [<bos>, x_1, \ldots, x_T]$ be an input sequence of length $T+1$ to the model. Consider a simple relative PE schema where dot product between query $\boldsymbol{q}_t$ and key $\boldsymbol{k}_i$ at positions $t$ and $i$ ($t \geq i$) can be expressed as: $\langle \boldsymbol{q}_t, \boldsymbol{k}_i \rangle := \boldsymbol{q}_t^T \boldsymbol{k}_i - (t-i)$. Then, there exists $\boldsymbol{W}_Q$, $\boldsymbol{W}_K$, $\boldsymbol{W}_V$, $\boldsymbol{W}_O$, $\boldsymbol{W}_1$, and $\boldsymbol{W}_2$ matrices, such that when $T < M$, $o_T > \mathcal{H}$; and when $T > M$, $o_T < \mathcal{H}$.*

Full proof is given in Appendix E.2. Theorems 3.1 and 3.2 state the failure of length extrapolation in NoPE and PE, respectively.

Building on Theorem 3.2, we further investigate the case for carefully orchestrated weave PE. The theoretical result is as follows:

> **Theorem 3.3** (Weave PE Extrapolation). *Let $N$ be a positive constant. Consider a simple weave PE extrapolation schema: when $t - i < N$, $\mathcal{W}(t - i) = t - i$; and when $t - i \geq N$, $\mathcal{W}(t - i) = N$. Then, the attention dot product is fixed as below:*
>
> $$\langle \boldsymbol{q}_t, \boldsymbol{k}_i \rangle := \begin{cases} \boldsymbol{q}_t^T \boldsymbol{k}_i - (t-i) & , \quad t - i < N \\ \boldsymbol{q}_t^T \boldsymbol{k}_i - N & , \quad t - i \geq N \end{cases}$$
>
> *, where $N \ll M$. Then, applying $\boldsymbol{W}_Q$, $\boldsymbol{W}_K$, $\boldsymbol{W}_V$, $\boldsymbol{W}_O$, $\boldsymbol{W}_1$, and $\boldsymbol{W}_2$ matrices from Theorem 3.2, we have when $T > M$, $o_T > \mathcal{H}$.*

Full proof is given in Appendix E.3. This theorem suggests that for existing LLMs relying on PE, simply weaving its relative positional encoding can effectively extend the input window.

Through Theorem 3.1 and Theorem 3.2, we formulate pertinent theoretical models for NoPE and PE, respectively, shedding light on the intricate relationship between extrapolation and the effective window length. Building upon these findings, Theorem 3.3 delves deeper into the realm of explicit PE, revealing that a well-designed weave PE scheme can effectively broaden the original effective window length. The theorems show the existence of an effective length $M$, which is typically related to the maximum training window length of the LLM. Furthermore, within the proof of Theorem 3.3, our results indicate that $N \ll M$, consistent to experimental findings in Su (2023b), where although the extrapolation position can theoretically start from 2048, the best extrapolation starting position is 512. More experimental parameters also validate this setting (refer to Appendix B.2).

### 3.4 Validating Extrapolation Using Observed Thresholds

Further, we design probe experiments (refer to Appendix F.3 for more results) to validate the observed phenomena and our theorems, as shown in Figure 2. From Figure 2, two observations are noted: Firstly, for hidden state values of the same dimension, the first layer undergoes minimal change, while the second layer exhibits a more pronounced transition. Exceeding the threshold implies extrapolation failure. This observation aligns with our theoretical model construction, where the first layer primarily refines positional information, with significant signal changes occurring in the second layer, as demonstrated in Theorem 3.2. Secondly, based on observational thresholds, when the length of the input sequence is around 12k, the values of hidden state corresponding to Dynamic-NTK Liu et al. (2023) surpass the threshold, implying extrapolation failure. Conversely, for ReRoPE Su (2023b), extrapolation succeeds. These two predictive outcomes corroborate with subsequent experimental results.

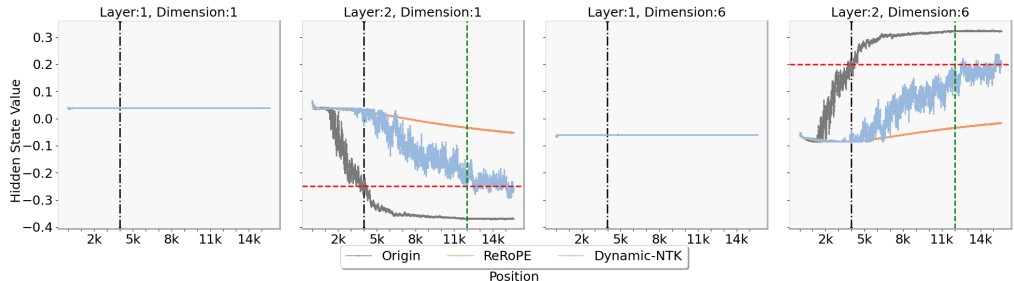

Figure 2: Thresholds for hidden states observed at specific dimensions on LLaMA2-7B-Chat, allowing for extrapolative judgments based on these thresholds. The vertical black dashed line indicate the position of maximum training length of the model. In this case, it is 4k for LLaMA2-7B-Chat model. The hidden state value at this position is designated as the observed threshold and marked with a horizontal red dashed line. When the hidden state value exceeds the red dashed line as the position changes, it signifies that the hidden state value has surpassed the threshold, suggesting a failure in extrapolation after that position.

## 4 Mesa-Extrapolation

In this section, we begin by introducing a novel weave position encoding method, termed Stair Position Encoding (Stair PE). Following this, we propose a chunk-based triangular attention matrix. Building on these concepts, we introduce the implementation of Mesa-Extrapolation. Lastly, we discuss the theoretical properties of these innovations.

### 4.1 Stair PE

Following the concept of weave PE in Equ.2, we define a novel weave PE method, namely Stair PE as follows:

$$\langle \boldsymbol{q}_t, \boldsymbol{k}_i \rangle := f_{StairPE}(q_t, k_i, t-i) = f_{PE}(q_t, k_i, \mathcal{W}(t-i)), \text{ and } \mathcal{W}(t-i) := \begin{cases} t-i & , & t-i \leq N \\ I & , & t-i > N \end{cases}$$

$$(3)$$

where $I = N + \left\lceil \frac{t-i-N}{E} \right\rceil$. $N$ denotes the extrapolated position, and $E$ denotes the extrapolated width. Both $N$ and $E$ are positive constants. Stair PE can be applied to existing relative PEs such as RoPE and ALiBi. For example, for RoPE:

$$\langle \boldsymbol{q}_t, \boldsymbol{k}_i \rangle := f_{\text{StairRoPE}}(\boldsymbol{q}_t, \boldsymbol{k}_i, t-i) = f_{\text{RoPE}}(\boldsymbol{q}_t, \boldsymbol{k}_i, \mathcal{W}(t-i)) = \boldsymbol{q}_t^T R^{\mathcal{W}(t-i)\theta} \boldsymbol{k}_i.$$

Taken a sequence of length 10 as an example, the right subplot of Figure 1 shows that the relative positions generated by PE (Both RoPE and ALiBi) are linear, while those generated by Stair PE (here $N = 4$ and $E = 2$) are non-linear. Since weave PE changes the linear relative position, it has to calculate the attention matrix more than once Su (2023b). Compared with ReRoPE (detailed on Appendix B.3), Stair PE provides a finer-grained extrapolated positions. Meanwhile, compared with Leaky-ReRoPE, Stair PE reuses known positions, reducing possible generalization errors. We also provide an ablation experiment to compare these weave PE methods (refer to Appendix C.6).

While our work was conducted independently, we note that Jin et al. (2024) have recently explored a similar idea through flooring the original positions and obtaining the relative position matrix with grouped self-attention. Although the two parallel thought processes are different, under certain conditions their formulations are equivalent (refer to Appendix G). Consequently, Self-Extend can be categorized as a Weave PE method. Our proposed Chunk-based Triangular Attention Matrix (detailed in Section 4.2) and its corresponding theoretical properties (Section 4.4) are also applicable to this parallel approach.

## 4.2 Chunk-based Triangular Attention Matrix

We design a chunk-based triangular attention matrix as shown in the left subplot of Figure 1. To achieve approximate linear memory consumption and computational speed, we further split the triangular attention matrix into several chunks and concatenate these chunks. We segment the input sequence into several sub-sequences according to *DynamicSplit* function (defined in Appendix 2), which divides a sequence into sub-sequences of equal length, with the exception of the first and last sequences. The length of each sub-sequence is determined by both the input token length and the max training length. Each of the generated sub-sequences then undergoes a self-attention operation to generate a corresponding chunk. That is, a sub-sequence of length $l$ will generate a corresponding chunk with the size $l \times l$.

## 4.3 Implementation

Mesa-Extrapolation mainly utilizes the chunk-based triangular attention matrix and Stair PE. Notice that regular PE (such as RoPE or ALiBi) is applied to all chunks except for the last chunk, for which Stair PE is applied. For the last chunk, all previous chunks are concatenated, and Stair PE is used to rearrange relative positional encoding to achieve extrapolation beyond the effective window length.

In summary, the process of Mesa-Extrapolation mainly contains four steps (Algorithm 1): The first three steps correspond to the *prefill* stage, which is used to calculate all input tokens. The last step corresponds to the *decoding* stage, which is used to generate next-token one by one.

Firstly, *DynamicSplit* function segments the input sequence, and the first segmented sub-sequence is fed into LLM to generate the first attention matrix chunk (line 3 in Algo.1). Secondly, subsequential sequences are iteratively processed while simultaneously feeding the key

---

**Algorithm 1** Mesa-Extrapolation Algorithm

**Require:** DynamicSplit, LLM, StairPE
**Input:** $s[0 : T-1]$ (input tokens with length T)
**Output:** $s[T, T+1, ...]$
**# Prefill Stage**
1: $first\_length, chunk\_width \leftarrow$ DynamicSplit($s$)
2: $K\_cache, V\_cache \leftarrow [], []$
3: $first\_K, first\_V \leftarrow$ LLM($s[0 : first\_length]$)
4: Append $first\_K$ to $K\_cache$, $first\_V$ to $V\_cache$
5: $i \leftarrow first\_length$
6: **while** $i < T - 1 - chunk\_width$ **do**
7: $K, V \leftarrow$ LLM($s[i : i + chunk\_width], first\_K, first\_V$)
8: $K\_cache$ append $K$, $V\_cache$ append $V$
9: $i \leftarrow i + chunk\_width$
10: **end while**
11: apply $StairPE$ to fix positions
12: $K, V \leftarrow$ LLM($s[i : T-1], K\_cache, V\_cache$)
**# Decoding Stage**
13: apply $StairPE$ to fix positions
14: generate next-token one by one

---

and value pairs of the first chunk into LLM to generate subsequent chunks (line 6-10 in Algo.1). Thirdly, the last sub-sequence is processed by concatenating the key and value pairs of all previous chunks together and using Stair PE to modify the relative positional encoding. Then, it is fed into

the LLM to produce the last chunk (line 11-12 in Algo.1). Finally, Stair PE is applied to process the current token and cached Key and Value pairs to generate next-token one by one (line 13-14 in Algo.1). We establish the effectiveness of our proposed Mesa-Extrapolation in the next section.

## 4.4 Theoretical Properties

Theorem 3.2 establishes a theoretical measurement for evaluating the effectiveness of extrapolation. We consistently apply this indicator, along with adjustments in relative positioning using Stair PE, to validate the feasibility of extrapolation. The result is as below:

**Corollary 4.1** (Mesa Extrapolation). *Let $N$ be a positive constant. Consider a simple Stair PE extrapolation schema, and the attention dot product is fixed as:*

$$\langle \boldsymbol{q}_t, \boldsymbol{k}_i \rangle := f_{\text{stairPE}}(\boldsymbol{q}_t, \boldsymbol{k}_i, t-i) = \begin{cases} \boldsymbol{q}_t^T \boldsymbol{k}_i - (t-i) & , \quad t-i < N \\ \boldsymbol{q}_t^T \boldsymbol{k}_i - I & , \quad t-i \geq N \end{cases}$$

*where $N \ll M$, $I = N + \lceil \frac{t-i-N}{E} \rceil$, and the extrapolated width $E$ is a constant. Then, Apply $\boldsymbol{W}_Q$, $\boldsymbol{W}_K$, $\boldsymbol{W}_V$, $\boldsymbol{W}_O$, $\boldsymbol{W}_1$, and $\boldsymbol{W}_2$ matrices from Theorem 3.2. Although $T > M$, it still $o_T > \mathcal{H}$.*

Full proof is provided in Appendix E.4. We prove that Mesa-Extrapolation can effectively extrapolate outside the max window length.

## 5 Experiments

In this section, we validate the performance of Mesa-Extrapolation through experiments measured over multiple metrics. We choose GovReport Huang et al. (2021), Pile Gao et al. (2020), LongBench Bai et al. (2023), and LongEval Krishna et al. (2023) datasets, and also generate a passkey dataset, which has been integrated in the code warehouse. More experimental details and results are referred to Appendix B and C.

Since our method is completely free plug-in and does not require fine-tuning, we choose methods of this type for comparison, including: model self (Origin), ReRoPE Su (2023b), Leaky-ReRoPE Su (2023b), Dynamic-NTK Liu et al. (2023), LM-Infinite Han et al. (2023), and Streaming-LLM Xiao et al. (2023).

We evaluate Mesa-Extrapolation using three prominent LLM families: LLaMA Touvron et al. (2023a) Touvron et al. (2023b) (including LLaMA-3B (Open-LLaMA-3B), LLaMA2-7B-Chat, and Vicuna-13B-V1.3), MPT Team (2023) (including MPT-7B), and PyThia Biderman et al. (2023) (including PyThia-6.9B and PyThia-12B). Notably, LLaMA and PyThia incorporate RoPE Su et al. (2023), whereas MPT employs ALiBi Press et al. (2021) – two of the most influential PE techniques in recent research. Furthermore, we validated our approach using the Phi-3-mini-128k-instruct model Microsoft (2024) (refer to Appendix C.9).

We also conduct ablation experiments (refer to Appendix C.6). We use a 2xA800 80GB NVIDIA GPU server as the experimental environment and adopt the PyTorch framework.

### 5.1 Evaluation on Passkey Retrieval Tasks

We assess the accuracy of Mesa-Extrapolation using the generated passkey dataset. This dataset comprises samples of varying lengths, each storing a random password at a random position. The sample length initiates at 1024 and increments by 1024. Simultaneously, 100 samples are randomly generated for each length. The proportion of correct answers found by LLMs is calculated for each input length.

Fig.3 shows the results of 6 LLMs on passkey retrieval task. The LLaMA families can employ various methods, including Origin, ReRoPE, Leaky-ReRoPE, Dynamic-NTK, LM-Infinite, Streaming-LLM, and our Mesa-Extrapolation. Note that these methods are model-specific and may not be universally applicable across all model series. For the MPT model, Origin, Streaming-LLM, ReRoPE, and Mesa-Extrapolation can be utilized. Similarly, for the PyThia model, Origin, Streaming-LLM, and Mesa-Extrapolation can be applied.

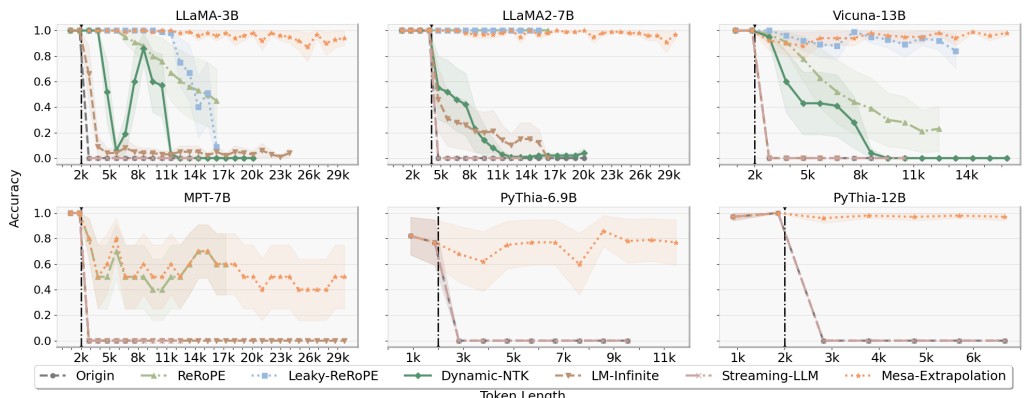

Figure 3: Passkey Retrieval Accuracy for different methods on various LLMs. X-axis represents the input token length, and Y-axis represents the accuracy of password found by LLMs. Different color regions denote the variance value, averaged on 100 samples for each input token length. The black dashed line represent the max training length for LLMs. Some observations: Weave PE-based methods, including ReRoPE, Leaky-ReRoPE, and Mesa-Extrapolation, consistently demonstrate stable extrapolation capabilities even when the input length surpasses the maximum training length. We claim that "early stopping" phenomenon in certain methods is attributed to GPU memory exhaustion under our existing hardware resources.

Analyzing the LLaMA model series reveals that weave PE-based methods, including ReRoPE, Leaky-ReRoPE and Mesa-Extrapolation, achieve superior extrapolation capabilities. Additionally, under our existing hardware constraints, Mesa-Extrapolation demonstrates longer extrapolation capabilities.

In the case of MPT model, after surpassing the maximum training length, the extrapolation capabilities of Mesa-Extrapolation and ReRoPE show a decline. We speculate that this may be attributed to an approximate ALiBi PE applied on MPT model. This approximate ALiBi would lead to significant disruptions when weaving its positions. (refer to Appendix C.2 for more details about MPT).

In the PyThia models, an additional observation is that 100% accuracy is still not achieved within the training length. We attribute this to the PyThia model being a pre-trained model without instruction alignment, resulting in a weakened intrinsic understanding of the task. Results with enhanced prompts are referred to Appendix C.5.

Dynamic-NTK exhibits partial extrapolation capabilities beyond the maximum training length. Streaming-LLM and LM-Infinite also exhibit poor performance. This is because these methods discard portions of the input token, leading to potential information loss and incorrect answers.

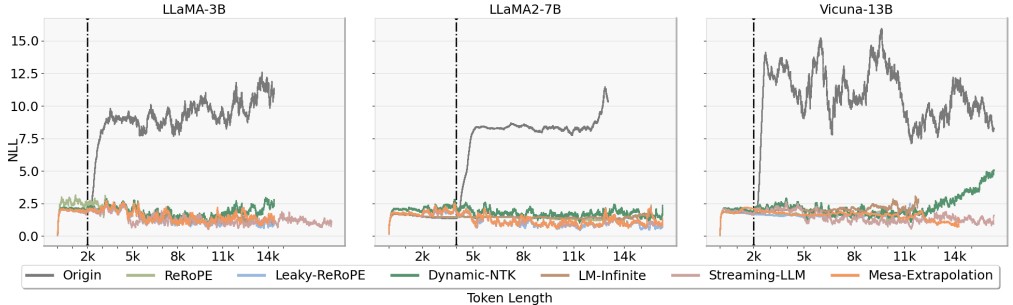

Figure 4: Perplexity (PPL) metrics on LLaMA models using the Pile dataset. Some observations: (1) The PPL value of Origin consistently increases when the maximum training length is exceeded. (2) Other methods maintain low PPL values, with Dynamic-NTK exhibiting a slight increase as the input length grows.

## 5.2 Evaluation on Language Modeling

We further assess the fluency of Mesa-Extrapolation utilizing the perplexity metric. Results evaluated on the Pile dataset are presented in Fig.4. X-axis represents the length of the input token, while the

Y-axis corresponds to NLL (Negative Log-Likelihood) values. It can be observed that the NLL value of Origin consistently increases when the maximum training length is exceeded. Other methods maintain low NLL values. LM-Infinite performs marginally better on Vicuna-13B. Dynamic-NTK method exhibits slightly weaker performance after 11k, and the performance continues to drop as the input length increases. In summary, our Mesa-Extrapolation demonstrates comparable performance to other methods on PPL metric.

## 5.3 Evaluation on Summary of Tasks

We conduct a summary task using the GovReport dataset and employ ROUGE ROUGE (2004) (ROUGE-1/2/L) as evaluation metrics. ROUGE assess overlapping N-grams by comparing the generated text with reference answers.

For the GovReport dataset, we segment the range from $3*1024$ to $11*1024$ based on sample length, with each interval of $1024$ units. A test set is created by randomly selecting $8$ samples from each interval. We choose LLaMA2-7B-Chat as the evaluated LLM. The experimental results for ROUGE is presented in Tables 1 below:

Table 1: ROUGE metric on LLaMA2-7B-Chat, averaged on $8$ samples within each interval using the GovReport dataset. Each cell contains ROUGE-1/ROUGE-2/ROUGE-L. The best values are marked in bold. Some observations: (1) Dynamic-NTK shows slightly better performance within 11k. (2) Other methods showcase the ability to achieve scores of varying degrees.

| Input Length | 3k | 4k | 5k | 6k | 7k | 8k | 9k | 10k | 11k |
|---|---|---|---|---|---|---|---|---|---|
| Origin | 32.5/11.4/29.7 | 36.6/14.7/33.9 | 9.4/1.9/8.6 | 1.6/0.1/1.6 | 0.0/0.0/0.0 | 0.0/0.0/0.0 | 0.0/0.0/0.0 | 0.0/0.0/0.0 | 0.0/0.0/0.0 |
| ReRoPE | 36.0/14.0/33.9 | 35.7/14.8/32.2 | 30.3/9.3/28.5 | 30.2/11.6/26.7 | **36.5**/12.6/34.2 | 31.1/10.0/28.3 | 35.7/13.3/33.0 | **35.7**/12.4/**32.5** | 34.6/13.0/32.5 |
| Leaky-ReRoPE | 33.6/12.3/30.9 | 36.3/14.3/33.5 | 35.7/**15.4**/32.7 | 25.4/7.4/23.6 | 31.3/13.8/29.2 | 34.1/12.1/30.3 | 30.8/9.7/27.9 | 30.9/11.8/28.2 | 30.8/10.8/28.1 |
| Dynamic-NTK | 38.2/13.9/36.1 | 39.5/16.0/36.0 | **35.9**/13.4/32.6 | 32.4/10.2/30.5 | 36.6/**15.9**/**34.9** | **35.2**/**13.1**/32.9 | **39.3**/**14.9**/**36.3** | 32.7/12.8/29.8 | **35.4**/**15.0**/32.7 |
| LM-Infinite | 34.6/11.8/31.8 | 35.2/15.1/32.6 | 33.5/12.6/30.7 | 31.1/10.4/29.6 | 31.4/12.9/29.0 | 32.1/10.8/28.2 | 27.0/8.4/24.1 | 27.9/9.1/25.6 | 29.4/9.8/25.9 |
| Streaming-LLM | **38.2**/**15.7**/**36.2** | 8.3/1.5/7.6 | 1.6/0.0/1.6 | 0.0/0.0/0.0 | 0.0/0.0/0.0 | 0.0/0.0/0.0 | 0.0/0.0/0.0 | 0.0/0.0/0.0 | 0.0/0.0/0.0 |
| Mesa-Extrapolation | 35.0/14.6/33.4 | **41.0**/**20.4**/**37.5** | 35.7/14.1/**34.1** | **33.6**/**12.8**/**31.4** | 34.1/11.4/31.6 | 32.7/10.9/30.4 | 30.7/11.3/27.8 | 33.9/**14.2**/30.4 | 30.2/10.1/26.6 |

In Table 1, we record the average scores of Rouge-1, Rouge-2, and Rouge-L within each interval. It is evident that once the effective input window is exceeded, the performance of Origin and Streaming-LLM declines rapidly, rendering it useless. For LM-Infinite, the scores exhibit a slight decrease as the length increases. Dynamic-NTK shows slightly better performance within 11k. However, combined with the Fluency experiment on Fig.4, it seems that the effective extrapolation range of Dynamic-NTK cannot exceed 12k, which is consistent with the threshold observed in Fig.2. Weave PE-based methods, including ReRoPE, Leaky-ReRoPE and Mesa-Extrapolation maintain similar generation quality as the length increases. Note our Mesa-Extrapolation shows slight variability in performance within mid-length (8k-11k) in the summary task. We speculate that with a fixed extrapolation width param ($E = 50$), the 7k-11k range may spread the model's attention more thinly compared to the 4k-6k range. We hypothesize that optimizing the extrapolation width could alleviate or improve performance in the 7k-11k range.

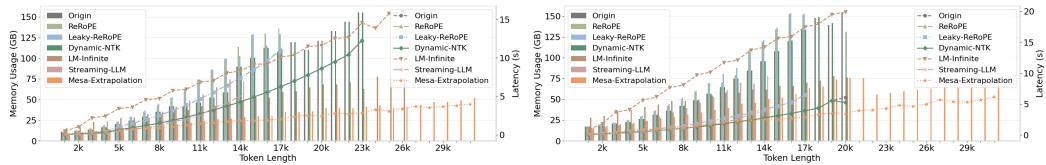

(a) Memory Usage & Latency for Open-LLaMA-3B  (b) Memory Usage & Latency for LLaMA2-7B

Figure 5: Memory Usage and Decoding Speed Comparison for LLaMA Models: 3B and 7B. The X-axis represents the input token length, the left Y-axis denotes memory usage, and the right Y-axis indicates speed about decoding time during inference. Some observations: (1) ReRoPE and Leaky-ReRoPE exhibit the largest memory footprint for the same input length, and their inference speed follows a quadratic function trend. (2) Mesa-Extrapolation shows an approximately linear inference speed, boasting the fastest inference speed and the smallest memory usage under the same input conditions.

## 5.4 Latency & Memory Usage

To compare actual memory consumption and inference speed, we conduct experiments using both the 3B and 7B versions of the LLaMA model. The results are presented in Fig.5. In Fig.5, the X-axis

represents the input token length, the left Y-axis denotes memory usage, and the right Y-axis indicates decoding time. It is noteworthy that decoding time is closely related to memory usage, primarily from the computation of the attention matrix.

Observing Fig.5, both memory usage and decoding time for Origin and Dynamic-NTK exhibit a quadratic trend. Similarly, ReRoPE and Leaky-ReRoPE exhibit the highest memory usage and decoding time, showcasing a quadratic trend, which aligns with our analysis (refer to Appendix C.7). Although LM-Infinite demonstrates a linear trend, its increase is substantial. In contrast, Mesa-Extrapolation method also exhibits a linear trend but significantly outperforms other methods in terms of memory usage and decoding time. Furthermore, as the input length increases, this trend becomes more pronounced.

## 6    Conclusion

Our study addresses the critical challenge faced by Large Language Models (LLMs) when confronted with longer input lengths, commonly referred to as the extrapolation problem. Through theoretical exploration, we uncover the underlying mechanisms of this challenge, shedding light on the reason of extrapolation failure for both NoPE and PE. Furthermore, we present theoretical evidence demonstrating the potential for effective extrapolation using Weave PE. Based on Weave PE, we introduce a practical solution called Mesa-Extrapolation, which strategically organizes input tokens into chunks to achieve competitive performance with minimal resource usage. Empirical validation demonstrates its effectiveness. Our work not only advances the understanding of the extrapolation problem but also offers a practical solution, with a complete free plug-in for LLMs.

**Limitations.** Mesa-Extrapolation is a plug-and-play method that does not require additional fine-tuning. However, previous work, "NTK-aware" ntk (2023) also shows that applying further fine-tuning to plug-in extrapolation is possible. Therefore exploring fine-tuning based on Mesa-Extrapolation can be an interesting next step. Due to limitations of resources, we have not yet validated our method at longer lengths.

**Broader Impacts** We contend that the significance of completely free plug-in extrapolation method lies in two aspects. Firstly, it enables the expansion of the effective window length of already trained LLMs with no additional cost. Secondly, it allows for training LLMs from scratch with short texts and subsequently expanding their effective window length with a free plug-in extrapolation method, which can greatly help the industry improve the extrapolation capabilities of diverse LLMs.

## 7    Acknowledgement

This work was supported by the National Key R&D Program of China under Grant No.2022ZD0160504 and Tsinghua University(AIR)-Asiainfo Technologies (China) Inc. Joint Research Center. We would also like to acknowledge anonymous reviewers and Zengxiang Lu for their valuable feedback and discussions.

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

# A    Related Work

**Extrapolation for PE**    A distinct line of research is dedicated to refining PE for enhanced extrapolation capabilities. Notable contributions include the introduction of Rotary Position Encoding (RoPE) Su et al. (2023), implementing relative PE through absolute position information. Similarly, Press et al. (2021) proposes a novel PE method *ALiBi*, fusing position information by directly introducing the relative position distance term in dot multiplication. In addition, there are other PE methods such as absolute position embedding (APE) Vaswani et al. (2017) and T5's Relative PE Raffel et al. (2020).

Another school of research focuses on enhancing the extrapolation performance of existing LLMs, categorized by whether to include further training. **The first subcategory** involves further fine-tuning, enlarging the effective window length by training LLMs on longer input texts. Chen et al. (2023) demonstrates that Position Interpolation (PI) method has a superior fine-tune effect, resulting in extended extrapolation capabilities with fewer fine-tuning steps. Mohtashami & Jaggi (2023) utilizes a new landmark token to represent individual input blocks, and enables model to select relevant blocks by further training, allowing for longer extrapolation. Zhang et al. (2024) follows similar idea by training a special token. Bertsch et al. (2023) proposes *Unlimiformer*, a k-nearest-neighbor (kNN) indexed encoder-decoder Transformer, enhancing efficiency by retrieving top-k keys for each decoder layer. Despite claiming support for decoder-only Transformer, differences in retrieval results across decoder layers may lead to potential failure. Tworkowski et al. (2023) introduces Focused Transformer (FOT), utilizing a contrastive learning-inspired training process to enhance the (key, value) space structure and enable effective context extension. **The second subcategory** explores methods that require no fine-tuning yet offer improved extrapolation through plug-ins. Su (2023b) introduces Rectified Rotary Position Embeddings (*ReRoPE*), a method that rectifies extrapolated relative positions based on RoPE within a specified interval to extend the effective window length. In addition, *Leaky-ReRoPE* offers an alternative by allowing a controlled leakage of position extrapolation within an interval. Both approaches are well-suited for LLaMA model families Touvron et al. (2023b) Touvron et al. (2023a), eliminating the need for fine-tuning. These methods do suffer from some drawbacks, such as twofold increase in memory consumption and longer inference time. We refer to this class of methods that achieve extrapolation by weaving the relative positions of PE without fine-tuning as *Weave PE*. We also notice that, recent work, InfLLM Xiao et al. (2024), proposes an additional memory units, which lookup token-relevant units for attention computation. In addition, it uses a modified encoding scheme similar to ReRoPE to achieve longer extrapolation. It is worth noting that the methods of the weave PE class can be applied seamlessly to these new designs. **However, there is still no theory to explain why the methods of weave PE class can work**. In a different line of inquiry, ntk (2023) proposes the "NTK-aware" method by drawing from the Neural Tangents (NTK) idea, which explores the high-frequency extrapolation and low-frequency interpolation concept for RoPE. Based on "NTK-aware", recent works bloc97 (2023), Peng et al. (2023), Roziere et al. (2023), Liu et al. (2023) perform further fine-tuning for optimal results.

Han et al. (2023) proposes LM-Infinite, which employs a $\Lambda$-shaped mask to prevent surpassing the effective window length by discarding central tokens. However, this design choice inevitably results in a loss of information. Likewise, Streaming-LLM Xiao et al. (2023) adopts a similar strategy to avoid surpassing the effective window length by discarding a portion of input tokens.

**Extrapolation for NoPE**    There also exists a counter perspective asserting that the position information of an input sequence can be perceived without utilizing PE. In a comprehensive experimental comparison presented in Kazemnejad et al. (2023), the decoder-only Transformer is shown to exhibit superior extrapolation properties with no position encoding (NoPE). Haviv et al. (2022) conjectures that causal attention enables the Transformer to infer position information without the help of PE, demonstrating its comparable performance with standard Transformer models through probing experiments. These new studies pose a key challenge regarding the choice of whether using PE or not in Transformer architecture.

**Theorems for Extrapolation**    We mainly focus on Transformer architecture. Although the Transformer architecture is considered as a "black box", there are ongoing efforts trying to explain its inner workings. Von Oswald et al. (2023) begins by providing a specific weight construction that elucidates the mechanistic understanding of in-context learning within optimized Transformers. Lindner et al. (2023) introduces interpretability for Transformers through programming, allowing the derivation of a program Transformer architecture tailored for specific tasks. In Han et al. (2023), the authors conclude that long-distance attention logits will explode. However, this does not clarify the

relationship between the effective window length and the existing supremum. It merely indicates an increase in the supremum, without directly implying an increase in the obtained actual attention logits. In Kazemnejad et al. (2023), inspired by programming Transformers Lindner et al. (2023), the authors construct a specific Transformer with predefined matrix parameters, theoretically demonstrating that NoPE can recover both absolute and relative position information even without the use of PE. However, these results fall short in explaining the underlying reasons for the failure of NoPE extrapolation. Built upon the entropy increase theory, Han et al. (2023) proves that with the expansion of input length, the entropy of attention experiences a corresponding increase. Additionally, Su (2021) introduces a scaling factor, denoted as $log_{\mathrm{train-len}}(n)$, which serves to mitigate entropy increase. Nevertheless, their theorems do not show that extrapolation fails beyond the effective window length.

We observe that extrapolation failure is intricately linked to the max window length. Nevertheless, existing theories on Transformers do not reveal the internal mechanism about extrapolation and the max window length. In light of this, our endeavor is to establish a correlation between these two aspects, aiming to offer theoretical insights that can guide us towards designing applicable extrapolation methodologies.

# B  Experiments Details

## B.1  Passkey Retrieval Dataset

The data used to perform the passkey task is constructed as follows:

TASK-DESCRIPT = "There is an important info hidden inside a lot of irrelevant text. Find it and memorize it. I will quiz you about the important information there."

DEFAULT-CONTENT = "The grass is green. The sky is blue. The sun is yellow. Here we go. There and back again."

KEY-CONTENT = "The pass key is {KEY}. Remember it. {KEY} is the pass key."

The TASK-DESCRIPT is placed at the beginning of the sample. Subsequently, the DEFAULT-CONTENT is repeated multiple times to serve as filler text. A randomly generated password is then created and inserted into the designated KEY-CONTENT section. Finally, a random position is selected, and the KEY-CONTENT, containing the generated password, is seamlessly incorporated into the sample.

The LLM is required to find the correct password from the sample.

## B.2  Params Setting

**Mesa-Extrapolation**  We design the DynamicSplit function used in Algorithm 1 to dynamically divide the input sequence into several subsequences according to the input length. The length of each subsequence corresponds to the size of the chunk. Let $\mathcal{C}$ denote the length of the each chunk (except the first chunk length $\mathcal{F}$ and the last chunk length $\mathcal{L}$). Define $\mathcal{I}$ as input token length and $\mathcal{T}$ as max training length.

The DynamicSplit function is detailed as follows:

In general, we set $\mathcal{F} = 100$, $\mathcal{M}_{max} = 200$ and $\mathcal{L} = 512$. Based on these criteria, we have devised a method for dynamically segmenting sequence. Leveraging the total length of the input token and the aforementioned requirements, we dynamically determine the length of each individual chunk.

Additionally, Stair PE is primarily employed in the manipulation of the last chunk. We need to concatenate all chunks together. After splicing chunks, the required positons will far exceed the max training length. At this time, we need to rearrange the positions according to Equ.3. In Equ.3, we generally set the extrapolated position $N = 512$ and set the extrapolated width $E = 50$.

**Implement Stair PE on RoPE and ALiBi**  The handling of RoPE Su et al. (2023) involves assigning positions to the query vector and key vector. The length of the query corresponds to the length of the last chunk, while the lengths of the key and value align with the combined lengths of all chunks. We assign position encoding to the query vector according the same sequence positions. Regarding the key's position encoding, we utilize the extrapolated width $E$ and extrapolated point $N$. By designing

**Algorithm 2** DynamicSplit function

**Require:** LLM parameters
**Input:** $s$ (input token sequence)
**Output:** $\mathcal{F}, \mathcal{C}$
1: set the values of $\mathcal{F}$ and $\mathcal{L}$ respectively as constants
2: set $\mathcal{T}$ according to the LLM parameters
3: get $\mathcal{I}$ according to input sequence $s$
4: $\mathcal{N} = \left\lfloor \frac{\mathcal{I} - \mathcal{L} - \mathcal{F}}{\mathcal{T} - \mathcal{F}} \right\rfloor$
5: $\mathcal{M} = (\mathcal{I} - \mathcal{L} - \mathcal{F}) \mod (\mathcal{T} - \mathcal{F})$
6: **if** $\mathcal{M} < \mathcal{M}_{max}$ **then**
7:    $\mathcal{C} = \mathcal{T} - \mathcal{F}$
8: **else**
9:    $\mathcal{C} = \left\lfloor \frac{\mathcal{I} - \mathcal{L} - \mathcal{F}}{\mathcal{N} + 1} \right\rfloor$
10: **end if**

the position encoding of query and key respectively, the relative position scheme defined by Stair PE can be implemented. And, for computational efficiency, we ensure that the relative PE of the last token in the last chunk completely follows Stair PE. The other tokens of the last chunk are similar to Stair PE.

For ALiBi's Press et al. (2021) position processing, we directly transform the relative position of the last chunk into the aforementioned structure within the mask matrix, refer to RoPE.

It is emphasized that the considerations outlined above primarily address the generation mechanism of LLM. The position encoding of the last chunk must align as closely as possible with the original position setting. This consideration is rooted in the belief that the latent concept Xie et al. (2021) formation heavily relies on there, with subsequent chunks providing essential information support. Experiments affirm that the last chunk significantly influences the accurate output of LLMs. This serves as the experimental foundation for our Mesa-Extrapolation method design.

**ReRoPE & Leaky-ReRoPE**  We follow Su (2023b) for configuring the parameters of ReRoPE and Leaky-ReRoPE. The extrapolated position for ReRoPE is established at 512, mirroring the setting for Leaky-ReRoPE. Simultaneously, the extrapolated increment for Leaky-ReRoPE is calculated in relation to the input length, following the formula below:

$$\frac{1}{k} = \frac{\mathcal{T} - w}{\mathcal{I} - w}$$

where $w$ is the extrpolated position as $512$, $\mathcal{T}$ is the maxium training length, and $\mathcal{I}$ is the input token length.

The ReRoPE and Leaky-ReRoPE methods are currently only applied to the LLaMA model series. We implemented ReRoPE based on its ideas on the MPT models.

**LM-Infinite & Streaming-LLM**  LM-Infinite Han et al. (2023) introduces two branches for attention masking: a global branch on the left and a local branch on the right. The global branch enables each token to attend to the preceding $n_{global}$ tokens if they appear before the current token. Conversely, the local branch allows each token to attend to preceding tokens within a distance of $n_{local}$. Tokens outside these two branches are disregarded during the attention operation. Subsequently, following its setting, set $n_{local} = \mathcal{I}$, where $\mathcal{I}$ still represents the training length. The choice of $n_{global}$ has a minor impact on model performance within the range $[10, 100]$, and thus, we fix $n_{global} = 100$ as its code setting. The distance limit entails the "effective distance" $d$ within $\mathcal{I}$, and we set $d = \mathcal{I}$.

Streaming-LLM Xiao et al. (2023) adopts a similar design to LM-Infinite. It introduces an initial token known as Attention Sinks, whose length is denoted by $x$. It also defines a Rolling KV Cache for retaining the most recent tokens, whose length is denoted by $y$. It set $x = 4$ and $y = \mathcal{T} - x$, where $\mathcal{T}$ denotes the maximum training length. The central tokens are referred to Evicted Tokens.

The distinguishing factor between the two approaches lies in the selection of the initial token length, with LM-Infinite opting to splice longer tokens. We speculate that this is also the reason why LM-Infinite performs better than Streaming-LLM in the experiments.

**Origin & Dynamic-NTK**   For Origin, it refers to loading the model directly, without using any methods. Dynamic-NTK differs from Origin by only adjusting the angle value of its RoPE component, according to the input length Liu et al. (2023).

The Dynamic-NTK method is currently applied to the LLaMA model series.

## B.3   Weave PE-based Schemes

We list the details of the weave PE-based methods, including ReRoPE and Leaky-ReRoPE, as follows:

$$
\text{ReRoPE=}\begin{bmatrix}
0 & & & & & & & & \\
1 & 0 & & & & & & & \\
2 & 1 & 0 & & & & & & \\
... & 2 & 1 & 0 & & & & & \\
N & ... & 2 & 1 & 0 & & & & \\
... & N & ... & 2 & 1 & 0 & & & \\
N & ... & N & ... & 2 & 1 & 0 & & \\
... & N & ... & N & ... & 2 & 1 & 0 & \\
N & ... & N & ... & N & ... & 2 & 1 & 0
\end{bmatrix}, \text{Leaky-ReRoPE} = \begin{bmatrix}
0 & & & & & & & & \\
1 & 0 & & & & & & & \\
2 & 1 & 0 & & & & & & \\
\vdots & 2 & 1 & 0 & & & & & \\
N & \cdots & 2 & 1 & 0 & & & & \\
N+\frac{1}{k} & N & \cdots & 2 & 1 & 0 & & & \\
\vdots & N+\frac{1}{k} & N & \cdots & 2 & 1 & 0 & & \\
N+\frac{L}{k} & \cdots & N+\frac{1}{k} & N & \cdots & 2 & 1 & 0
\end{bmatrix}
$$

where left ReRoPE shows relative position on attention matrix, including (1) Numbers marked on each cell denote its relative position $(t-i)$ between $q_t$ and $k_i$. (2) It start to extrapolate position from relative position $N$. And right Leaky-ReRoPE shows relative position on attention matrix, including (1) Numbers marked on each cell denote its relative position $(t-i)$ between $q_t$ and $k_i$. (2) It start to extrapolate position from relative position $N$, with an incremental factor $\frac{1}{k}$.

With Stair PE defined on Equ.3, the extrapolated position after $N$ exhibits a fixed extrapolated width $E$, resembling a stair-like increment. Our Mesa-Extrapolation is designed with a finer position granularity compared to ReRoPE. Simultaneously, unlike Leaky-ReRoPE, ours utilizes previously trained positions, ensuring greater stability.

While these weave PE schemes, including ReRoPE, Leaky-ReRoPE, and Stair PE, demonstrate theoretical feasibility, their practical implementation may encounter computational complexities. For instance, both ReRoPE and Leaky-ReRoPE are primarily employed in the LLaMA models, which is based on RoPE Su et al. (2023). Due to the distinctive nature of RoPE, these methods necessitate the calculation of the attention matrix more than once, resulting in double the normal memory consumption. Moreover, given the quadratic complexity of the calculations, as the input length expands, both inference time and memory consumption grow proportionally.

## B.4   ALiBi on MPT-7B

It is worth noting that the ALiBi implementation of MPT is only an approximation, as follow:

$$
\begin{bmatrix}
-9 & & & & & & & & & \\
-9 & -8 & & & & & & & & \\
-9 & -8 & -7 & & & & & & & \\
-9 & -8 & -7 & -6 & & & & & & \\
-9 & -8 & -7 & -6 & -5 & & & & & \\
-9 & -8 & -7 & -6 & -5 & -4 & & & & \\
-9 & -8 & -7 & -6 & -5 & -4 & -3 & & & \\
-9 & -8 & -7 & -6 & -5 & -4 & -3 & -2 & & \\
-9 & -8 & -7 & -6 & -5 & -4 & -3 & -2 & -1 & \\
-9 & -8 & -7 & -6 & -5 & -4 & -3 & -2 & -1 & 0
\end{bmatrix}
$$

where showcase an input token length 10 for ALiBi PE mask. Based on this approximate implementation of ALiBi, we still use splitting chunk and Stair PE to implement Mesa-Extrapolation. We speculate that the approximation of ALiBi is the reasons for the instability of the extrapolation on Accuracy.

Table 2: BLEU metric for mean and standard variance on LLaMA2-7B-Chat, averaged on 8 samples within each interval using the GovReport dataset. The best values are marked in bold. Some observations: (1) Dynamic-NTK shows slightly better performance within 11k. (2) Weave PE-based methods showcase the ability to achieve scores of varying degrees.

| model-name | 1k | 2k | 3k | 4k | 5k | 6k | 7k | 8k | 9k | 10k | 11k |
|---|---|---|---|---|---|---|---|---|---|---|---|
| Origin | **0.158** ± 0.044 | 0.118 ± 0.058 | 0.049 ± 0.043 | 0.063 ± 0.033 | 0.008 ± 0.015 | 0.0 ± 0.0 | 0.0 ± 0.0 | 0.0 ± 0.0 | 0.0 ± 0.0 | 0.0 ± 0.0 | 0.0 ± 0.0 |
| ReRoPE | 0.117 ± 0.032 | 0.047 ± 0.022 | 0.053 ± 0.036 | 0.067 ± 0.039 | 0.022 ± 0.014 | 0.028 ± 0.015 | 0.041 ± 0.02 | 0.034 ± 0.02 | 0.032 ± 0.014 | 0.046 ± 0.019 | 0.074 ± 0.041 |
| Leaky-ReRoPE | 0.046 ± 0.041 | 0.105 ± 0.071 | 0.05 ± 0.041 | 0.049 ± 0.035 | 0.055 ± 0.035 | 0.011 ± 0.012 | 0.055 ± 0.036 | 0.032 ± 0.026 | 0.029 ± 0.02 | 0.051 ± 0.043 | 0.036 ± 0.047 |
| Dynamic-NTK | 0.075 ± 0.035 | 0.082 ± 0.043 | 0.069 ± 0.025 | **0.086** ± 0.038 | **0.068** ± 0.022 | 0.052 ± 0.01 | **0.078** ± 0.024 | **0.057** ± 0.022 | **0.086** ± 0.031 | 0.07 ± 0.024 | **0.077** ± 0.013 |
| LM-Infinite | 0.158 ± 0.051 | **0.119** ± 0.058 | 0.039 ± 0.041 | 0.069 ± 0.055 | 0.054 ± 0.037 | 0.031 ± 0.026 | 0.029 ± 0.011 | 0.032 ± 0.017 | 0.023 ± 0.029 | 0.024 ± 0.02 | 0.032 ± 0.027 |
| Streaming-LLM | 0.136 ± 0.026 | 0.037 ± 0.028 | **0.072** ± 0.041 | 0.004 ± 0.009 | 0.0 ± 0.0 | 0.0 ± 0.0 | 0.0 ± 0.0 | 0.0 ± 0.0 | 0.0 ± 0.0 | 0.0 ± 0.0 | 0.0 ± 0.0 |
| Mesa-Extrapolation | 0.14 ± 0.035 | 0.041 ± 0.015 | 0.059 ± 0.053 | 0.053 ± 0.037 | 0.036 ± 0.017 | **0.055** ± 0.033 | 0.063 ± 0.03 | 0.040 ± 0.023 | 0.055 ± 0.033 | **0.074** ± 0.020 | 0.045 ± 0.027 |

## C  More Experimental Results

### C.1  BLEU Results on GovReport

We use the GovReport dataset and employ BLEU Papineni et al. (2002) as evaluation metric, with a same setting applied on Table 1. The experimental result for BLEU is presented in Table 2. In Table 2, we present the mean and standard variance for each method across various interval lengths. It is evident that, as the input token length increases, the performance consistently deteriorates for LM-Infinite. Streaming-LLM also experiences extrapolation failure beyond the model's inherent effective window length of 4k.

Dynamic-NTK shows slightly better performance within 11k. Here, the BLEU scores exhibit almost consistent results with those of ROUGE on Table 1. It is also noteworthy that additional experiments show performance degradation of Dynamic-NTK beyond 11k, likely suggesting a limited effective extrapolation range for Dynamic-NTK.

Weave PE-based methods, including ReRoPE, Leaky-ReRoPE, and Mesa-Extrapolation, consistently demonstrates competitive results, maintaining similar generation quality as the length increases.

Overall, despite the chunking applied to our Mesa-Extrapolation method, it maintains a competitive edge compared to ReRoPE and Leaky-ReRoPE. This is achieved even with the trade-off between losing a certain amount of information and reducing memory consumption and inference time.

The results of Mesa-Extrapolation in the summary task, focusing on the generation of summary text within 1000 tokens across different input lengths, are detailed in Table 5.

### C.2  Perplexity (PPL) metrics on MPT-7b

Following the same metrics setting about LLaMA models on Fig.4, we plot the NLL result about MPT-7B on Fig.6 as below:

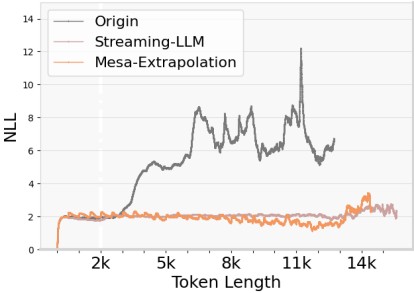

Figure 6: PPL Metric Comparison for MPT-7B using Origin, Streaming-LLM, and Mesa-Extrapolation. The white dashed line represents MPT-7B's maximum training length at 2k. Some observations: (1) For Origin, after surpassing the maximum training length, extrapolation extends to approximately 3.5k, leading to a rapid escalation in PPL. (2) Both Mesa-Extrapolation and Streaming-LLM models exhibit effective negative log-likelihood (NLL) values.

In Fig.6, both Mesa-Extrapolation and Streaming-LLM demonstrate effective negative log-likelihood (NLL) values. For the original MPT model, after the extrapolation is extended 3.5k, it will cause the perplexity (PPL) level to rise rapidly.

ALiBi Press et al. (2021) can still effectively extrapolate to around 3.5k position after surpassing the maximum training length of 2k. This observation affirms the extrapolation capability of the ALiBi method, especially when compared with RoPE's extrapolation results.

However, it is essential to highlight that while ALiBi achieves extrapolation to 3.5k in above NLL task, its performance in the Passkey and LongEval tasks (refer to C.3) reveals limitations. This discrepancy is attributed to an inherent characteristic of ALiBi. ALiBi's expression (refer to D.2) indicates a constant decrease in the value of its position encoding as the relative distance increases. Considering the $\mathrm{SoftMax}$ operation within the attention mechanism, under identical conditions, the attention score obtained becomes smaller. This implies that tokens situated farther away receive less attention, possibly leading to the neglect of these distant tokens. Therefore, despite ALiBi's apparent success in effectively extrapolating the NLL task to 3.5k, it falls short in attending to tokens beyond the maximum training length in practice. A similar analysis process is as Su (2023a).

It's noted that, NoPE Kazemnejad et al. (2023) claims to possess extrapolation capabilities akin to ALiBi. We hypothesize that the underlying reasons for NoPE's extrapolation performance align with the analysis presented here.

## C.3    Evaluation on LongEval

We conduct additional testing on LongEval Krishna et al. (2023) lines task, a recently prominent evaluation task for long texts.

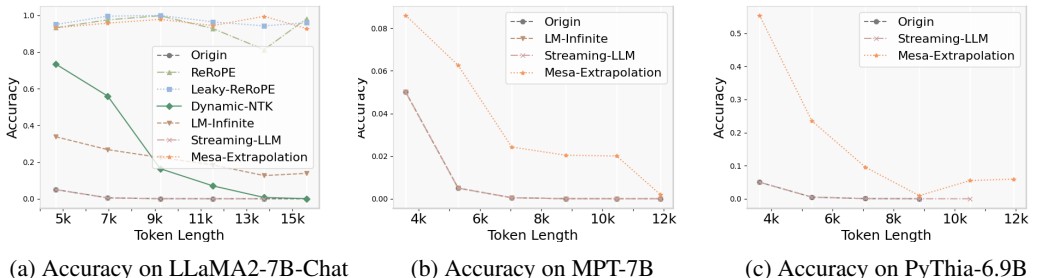

(a) Accuracy on LLaMA2-7B-Chat    (b) Accuracy on MPT-7B    (c) Accuracy on PyThia-6.9B

Figure 7: LongEval Lines Task on LLMs using various methods. Some observations: (1) Origin and Streaming-LLM consistently exhibit an inability to extrapolate beyond the effective window length. (2) LM-Infinite shows weak extrapolation ability on the LLaMA2-7B-Chat model and fails to extrapolate on the MPT-7B model. (3) Weave PE-based methods, including ReRoPE, Leaky-ReRoPE, and Mesa-Extrapolation, all exhibit commendable extrapolation capabilities on the LLaMA2-7B-Chat model. (4) For the MPT-7B model, all methods display weak extrapolation performance, with Mesa-Extrapolation slightly outperforming other methods. (5) Mesa-Extrapolation also shows a certain extrapolation ability on the PyThia-6.9B model, with a decrease in extrapolation performance as the input length increases.

Fig.7 show the accuracy of LongEval Lines Task. Origin and Streaming-LLM consistently exhibit an inability to extrapolate beyond the effective window length. LM-Infinite shows weak extrapolation ability on the LLaMA2-7B-Chat model and fails to extrapolate on the MPT-7B model. Dynamic-NTK method, as shown earlier, almost fails after the input token length exceeds 11k. Weave PE-based methods, including ReRoPE, Leaky-ReRoPE, and Mesa-Extrapolation, all exhibit commendable extrapolation capabilities on the LLaMA2-7B-Chat model. For the MPT-7B model, all methods display weak extrapolation performance, with Mesa-Extrapolation slightly outperforming other methods. We analyze that the reason for the failure of extrapolation in the MPT-7B model is attributed to the approximated implementation of ALiBi PE. This approximation makes it susceptible to interference when use weave PE (refer to C.2).

Mesa-Extrapolation also shows a certain extrapolation ability on the PyThia-6.9B model, with a decrease in extrapolation performance as the input length increases.

## C.4 Evaluation on LongBench

We select LongBench Bai et al. (2023) dataset and use 5 major categories of tasks, including Single-Document QA, Multi-Document QA, Few-shot Learning, Synthesis Tasks and Code Completion. Among them, each task selects a dataset, namely qasper, hotpotqa, samsum, passage-retrieval-en, and repobench-p. Taking into account the varying memory consumption of different methods, we opt to filter out samples with an input length exceeding 10k to prevent discrepancies caused by out-of-memory (OOM) issues. We use the abbreviations in Table 3: S-Document (Single Document) QA, M-Document (Multi Document) QA, F-Learning (Few-shot Learning), S-Tasks (Synthesis Tasks), C-Completion (Code Completion).

Table 3: Accuracy on LongBench across multiple tasks using LLaMA2-7B-Chat. Some observations: (1) Dynamic-NTK shows good performance, especially for Code Completion. (2) LM-Infinite shows slightly weaker performance. (3) Mesa-Extrapolation shows better performance on most tasks.

| Multi-Tasks | S-Document QA | | M-Document QA | | F-Learning | | S-Tasks | | C-Completion | |
|---|---|---|---|---|---|---|---|---|---|---|
| Input Token Length | 4-8k | 8k+ | 4-8k | 8k+ | 4-8k | 8k+ | 4-8k | 8k+ | 4-8k | 8k+ |
| Origin | 1.3 | 0 | 1.25 | 0 | 3.03 | 0 | 2.33 | 0 | 4.72 | 1.11 |
| Dynamic-NTK | 22.22 | 11.7 | 35.2 | 19.4 | 37.96 | 38.32 | 12.4 | **6.06** | **34.28** | **50.14** |
| LM-Infinite | 17.39 | 12.22 | 32.52 | 30.55 | 38.25 | 36.17 | 10.85 | 3.03 | 33.09 | 43.89 |
| Streaming-LLM | 1.22 | 0 | 1.25 | 0 | 3.05 | 0 | 2.33 | 0 | 5.68 | 0.21 |
| Mesa-Extrapolation | **24.69** | **20.24** | **36.72** | **42.76** | **38.63** | **38.41** | **14.73** | 3.03 | 20.6 | 22.39 |

In Table 3, Origin and Streaming-LLM barely work after exceeding the maximum training length of 4k. LM-Infinite shows slightly weaker performance. Dynamic-NTK shows good performance, especially for Code Completion. Considering the effective scope of Dynamic-NTK, we speculate that this is related to the setting within 10k length. Mesa-Extrapolation shows better performance on most tasks.

## C.5 Enhanced Prompts for Mesa-Extrapolation

Considering that Mesa-Extrapolation involves positional approximation, as the input length increases, attention becomes more dispersed, resulting in increased entropy. One possible approach to mitigate these entropy increases is to utilize instruction-aligned prompts. These prompts may represent the strongest latent concepts that the model can follow, while clearly articulating the goals to be achieved. Similarly, it may be the reason why PyThia cannot achieve 100% accuracy within its training length on Fig.3, due to a lack of instruction alignment.

Based on these considerations, we employ the model's corresponding prompt Geng, Xinyang and Liu, Hao (2023) and augment it with explicit goal statements, referred to as an enhanced prompt, like below:

```
Q: what is the passkey in below text?  + Sample \n A:
```

or

```
Q: Sample \n A: the passkey is
```

Additionally, we leverage model parallel inference technology BlackSamorez (2023) and perform verification concurrently on 2 A800 GPUs.

Fig.8 show the accuracy on 3 LLaMA models with enhanced prompts using Mesa-Extrapolation. We declare that the longest input length that these 3 models can currently reach is the limit of our existing hardware resources. For LLaMA-3B, the enhanced prompts improve the Accuracy and is extrapolated to 60k. (2) Both of LLaMA2-7B-Chat and Vicuna-13B models, show good improvements by the enhanced prompts.

Through this experiment, we hypothesize that by writing prompts more carefully and accurately, we can maximize the extrapolation performance of free plug-in.

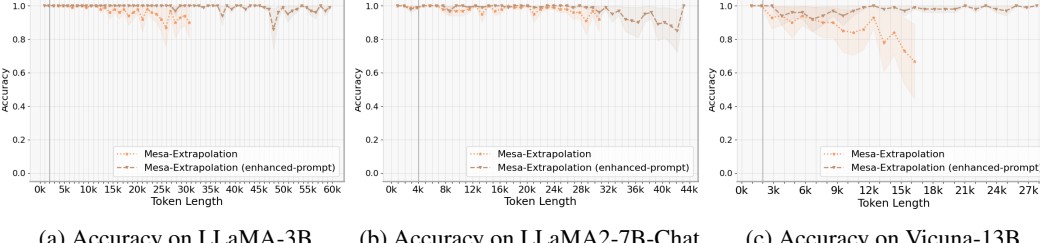

| (a) Accuracy on LLaMA-3B | (b) Accuracy on LLaMA2-7B-Chat | (c) Accuracy on Vicuna-13B |

Figure 8: Accuracy about Passkey Retrieval Tasks using enhanced prompts for Mesa-Extrapolation. The gray line represents the max training length at 2k, 4k, and 2k, respectively. Some observations: (1) On the LLaMA-3B model, the enhanced prompts significantly improves the Accuracy and is extrapolated to 60k. (2) Both of LLaMA2-7B-Chat and Vicuna-13B models, show good improvements by the enhanced prompts.

## C.6 Ablation Experiments

We design ablation experiment about weave PE based methods. We apply the encoding schemes of ReRoPE and Leaky-ReRoPE to chunk-based triangular attention matrices, aligning them with our implemented Mesa-Extrapolation. The entire design of the ablation experiment involved controlling only one variable, the weaving PE scheme, applying different weaving PE methods to the processing of the last chunk, while using the same dataset and testing environment. The experimental results 9 are as follows:

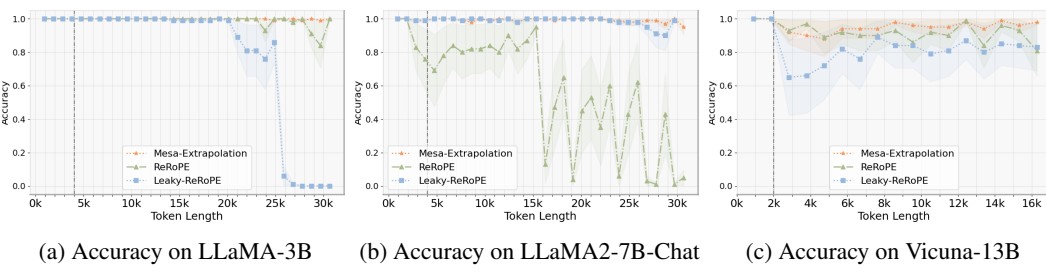

| (a) Accuracy on LLaMA-3B | (b) Accuracy on LLaMA2-7B-Chat | (c) Accuracy on Vicuna-13B |

Figure 9: Accuracy about Passkey Retrieval Tasks using chunk-based triangular attention matrix.

Figure 9 demonstrate that both ReRoPE and Leaky-ReRoPE experience a certain loss in accuracy as the input sequence grows. We speculate that this may be due to ReRoPE repeatedly using the same extrapolation positions, while Leaky-ReRoPE, employing some fractions, may not be as precise as Stair PE in comparison to normal relative positions, resulting in a slight decrease in effectiveness.

## C.7 Theoritical Speed & Memory

We assess the computational memory usage and inference time about decoding speed on various methods.

Table 4: Theoretical Memory Usage Based on Attention Matrix for Different Methods. Observations: (1) Origin and Dynamic-NTK exhibit identical quadratic memory consumption. (2) ReRoPE and Leaky-ReRoPE demonstrate $2\times$ the memory consumption of Origin. (3) LM-Infinite, Streaming-LLM, and Mesa-Extrapolation showcase linear memory consumption.

| Methods | Origin | ReRoPE | Leaky-ReRoPE | Dynamic-NTK | LM-Infinite | Streaming-LLM | Mesa-Extrapolation |
|---|---|---|---|---|---|---|---|
| Memory | $\mathcal{O}(n^2)$ | $2\times\mathcal{O}(n^2)$ | $2\times\mathcal{O}(n^2)$ | $\mathcal{O}(n^2)$ | $\mathcal{O}((1+\sqrt{2})n)$ | $\mathcal{O}((1+\sqrt{2})n)$ | $\mathcal{O}((2+\sqrt{2})n)$ |

Table 4 shows the theoretical result about the memory usage. For Origin, the attention matrix calculation requires the current token to compute the attention score with each previous token, resulting in a memory footprint of $\mathcal{O}(n^2)$. Dynamic-NTK only alters the angle base, making it different from the Origin but still quadratic. For ReRoPE and Leaky-ReRoPE, as showed in Su (2023b), since the attention matrix needs to be calculated twice, their memory footprint is $2 \times \mathcal{O}(n^2)$. For Mesa-Extrapolation, its strategy involves splitting chunks while avoiding the

quadratic term. Accounting for splicing in the last chunk, its total memory usage scales proportional to $\mathcal{O}((2 + \sqrt{2})n)$. LM-Infinite and Streaming-LLM adopt a similar $\Lambda$-shaped mask, that scales proportional to $\mathcal{O}((1 + \sqrt{2})n)$.

## C.8 Generated Summary using Mesa-Extrapolation

We present here the results of the summary generation task using GovReport Huang et al. (2021). In this experiment, task is to generate a summary for texts of varying lengths, limited to 1000 tokens. The outcomes yielded by Mesa-Extrapolation are outlined below on Table 5:

Table 5: Mesa-Extrapolation undertakes summarization tasks and generates text based on varying input lengths from the GovReport dataset. Some observations: (1) Mesa-Extrapolation adeptly and systematically summarizes key points one by one across various input lengths. It showcases logical coherence and demonstrates commendable summarization capabilities. (2) Additionally, Mesa-Extrapolation accurately incorporates punctuation and exhibits an automatic stopping mechanism for proper stopping.

| Input length | Mesa-Extrapolation generate summary texts within 1000 tokens |
|---|---|
| 5k | The Commercial Crew Program of NASA awarded two contracts in 2014 to Boeing and SpaceX, valued at up to $4.2 billion, to develop and demonstrate crew transportation systems that meet NASA requirements for the International Space Station (ISS). The program is experiencing delays and schedule uncertainty due to technical issues and safety concerns with the contractors' systems. NASA has been working to address these issues and has made progress in testing and evaluating the contractors' systems, but there are still risks and uncertainties associated with the program. The main content of the report includes: 
 * NASA awarded two contracts to Boeing and SpaceX in 2014 to develop and demonstrate crew transportation systems for the ISS. 
 * The program is experiencing delays and schedule uncertainty due to technical issues and safety concerns with the contractors' systems. 
 * NASA has been working to address these issues and has made progress in testing and evaluating the contractors' systems, but there are still risks and uncertainties associated with the program. 
 * The program's ability to process certification paperwork is creating uncertainty about the timing of certification for either contractor, which could result in delays to the first operational mission to the ISS. 
 * NASA has announced plans to purchase two additional Soyuz seats from Russia to extend U.S. access to the ISS beyond September 2020, if delays to certification of the Commercial Crew Program contractors continue. 
 * The report highlights the need for NASA to maintain a contingency plan to ensure uninterrupted access to the ISS if delays with the Commercial Crew Program contractors continue beyond September 2020. |

| Input length | Mesa-Extrapolation generate summary texts within 1000 tokens |
|---|---|
| 6k | The main content of the report is about the contract financing policies and practices of the Department of Defense (DOD) and their impact on the defense industry. The report highlights the following key points:
* DOD has long recognized that contract financing is capital-intensive and takes a long time to complete, and therefore provides contract financing to help contractors manage their expenses.
* DOD provides contract financing through progress payments and performance-based payments. Progress payments are based on the percentage of work completed, while performance-based payments are based on the contractor's achievement of specific milestones.
* The Federal Acquisition Streamlining Act of 1994 requires DOD to use performance-based payments whenever practicable. However, the use of performance-based payments has decreased since the act was passed.
* In 2010, DOD issued a proposed rule to amend the Defense Federal Acquisition Regulation Supplement (DFARS) to provide for performance-based payments. The proposed rule would have allowed contractors to receive payments based on their performance, but the rule was later withdrawn.
* Despite the decreased use of performance-based payments, DOD has not conducted a comprehensive assessment of the impact of its contract financing policies on the defense industry in over 30 years.
* GAO recommends that the Acting Secretary of Defense direct the Under Secretary of Defense for Acquisition and Sustainment to conduct a comprehensive assessment of the effect of DOD's contract financing policies on the defense industry and update the assessment on a recurring basis.
* The report also notes that changes in the defense industry, such as the emergence of new entrants and changes in market and economic conditions, have affected the defense industry's ability to attract new entrants.
* DOD has acknowledged the need to assess contract financing policies against market conditions on an ongoing basis, but has not done so comprehensively in over 30 years.
* The report highlights the need for DOD to update its contract financing policies to reflect changes in the defense industry and to ensure that they are effective in supporting the defense industry. |

| Input length | Mesa-Extrapolation generate summary texts within 1000 tokens |
|---|---|
| 7k | The report summarizes the results of a review of the monitoring and evaluation (M&E) policies of six US government agencies that administer foreign assistance: the Department of Defense (DOD), the Department of State (State), the Millennium Challenge Corporation (MCC), the US Agency for International Development (USAID), the Department of Health and Human Services (HHS), and the Department of Agriculture (USDA). The review was conducted in response to the Foreign Aid Transparency and Accountability Act of 2016, which requires the Office of Management and Budget (OMB) to issue guidelines for M&E of foreign assistance. The report finds that while the agencies have taken steps to implement the guidelines, there are some areas where they could improve. For example: 
 * DOD, HHS, and USDA did not include the requirement to conduct impact evaluations for all pilot programs, which could lead to duplication or scaling up of programs without fully understanding their effectiveness. 
 * State, USAID, and MCC have policies that generally address leading practices for M&E, but some agencies have not incorporated all of the required practices into their policies. 
 * OMB's guidelines do not include leading practices for program evaluation, which could limit the usefulness of foreign assistance program evaluations. 
 * Some agencies have not established roles and responsibilities for evaluation activities when funds are transferred between or among agencies, which could result in a lack of accountability for evaluation activities. 
 * The guidelines do not require agencies to develop monitoring plans based on risk assessments, which could result in a lack of focus on high-risk programs. 
 * The guidelines do not require agencies to provide justifications for not conducting impact evaluations, which could result in a lack of transparency and accountability. 
 * The guidelines do not specify requirements for evaluating the effectiveness of pilot programs, which could result in a lack of understanding of the impact of these programs. 
 * The guidelines do not require agencies to develop staff skills in evaluation and use of evidence, which could result in a lack of capacity to conduct effective evaluations. 
 * The guidelines do not provide specific requirements for evaluating the effectiveness of interagency agreements, which could result in a lack of accountability for these agreements. 
 * The guidelines do not require agencies to provide timely evaluation results to decision-makers, which could result in a lack of usefulness of evaluation findings. 
 The report makes seven recommendations for improvement, including that OMB should update the guidelines to include leading practices for program evaluation, and that agencies should establish roles and responsibilities for evaluation activities when funds are transferred between or among agencies. The report also recommends that agencies should develop monitoring plans based on risk assessments, provide justifications for not conducting impact evaluations, and evaluate the effectiveness of pilot programs. |

| Input length | Mesa-Extrapolation generate summary texts within 1000 tokens |
| --- | --- |
| 8k | The main content of the report is focused on the challenges and limitations faced by dual-eligible beneficiaries in the Medicare and Medicaid programs, particularly in the coordination of care and the alignment of enrollment. The report highlights the following key findings:
1. Difficulty in accessing care: Dual-eligible beneficiaries face challenges in accessing care due to the complexity of the Medicare and Medicaid systems, leading to delays in receiving necessary care.
2. Limited information on experiences: There is a lack of quality information on the experiences of dual-eligible beneficiaries who have been default enrolled into D-SNPs, which hinders CMS's ability to identify the extent of the challenges faced by these beneficiaries and determine how to address them.
3. Limited coordination of care: Dual-eligible beneficiaries may experience limited coordination of care due to the lack of alignment between Medicare and Medicaid systems, leading to fragmented care and confusion among beneficiaries.
4. Limited provider networks: Dual-eligible beneficiaries may face limited provider networks, which can make it difficult for them to access necessary care.
5. Limited enrollment options: Dual-eligible beneficiaries may have limited enrollment options, which can lead to default enrollment into D-SNPs without their knowledge or consent.
6. Limited quality information: There is a lack of quality information on the experiences of dual-eligible beneficiaries who have been default enrolled into D-SNPs, which hinders CMS's ability to identify the extent of the challenges faced by these beneficiaries and determine how to address them.
7. Limited alignment of enrollment: Dual-eligible beneficiaries may face limited alignment of enrollment between Medicare and Medicaid, leading to confusion and fragmented care.
8. Limited provider networks: Dual-eligible beneficiaries may face limited provider networks, which can make it difficult for them to access necessary care.
9. Limited enrollment options: Dual-eligible beneficiaries may have limited enrollment options, which can lead to default enrollment into D-SNPs without their knowledge or consent.
10. Limited quality information: There is a lack of quality information on the experiences of dual-eligible beneficiaries who have been default enrolled into D-SNPs, which hinders CMS's ability to identify the extent of the challenges faced by these beneficiaries and determine how to address them.
The report highlights the need for CMS to take steps to address these challenges and improve the coordination of care and alignment of enrollment for dual-eligible beneficiaries. The report recommends that CMS obtain quality information on the experiences of dual-eligible beneficiaries who have been default enrolled into D-SNPs and take steps to improve the alignment of enrollment between Medicare and Medicaid. |

Table 5 – *Continued from previous page*

| Input length | Mesa-Extrapolation generate summary texts within 1000 tokens |
|---|---|
| 9k | The report summarizes the actions taken by the Army to improve its marketing and advertising program in response to recommendations from the U.S. Army Audit Agency (AAA) in its April 2018 reports. The Army has taken several steps to address the recommendations, including:
1. Revising strategic goals: The Army has revised its strategic goals to focus on the activation phase of the consumer journey, which includes providing potential recruits with information about the Army's opportunities and benefits.
2. Updating and documenting its assessment process: The Army has updated its assessment process to include more detailed information on the effectiveness of its marketing and advertising efforts.
3. Improving data systems: The Army has taken steps to improve its data systems, including developing a new resource requirements projection model and updating its marketing and advertising regulation.
4. Integrating national and local marketing and advertising efforts: The Army has established a new office, the Army Marketing and Research Group (AMRG), to coordinate and oversee its marketing and advertising efforts at the national and local levels.
5. Obtaining new tools to determine required marketing and advertising resources: The Army has contracted with a primary advertising agency to develop and implement a marketing and advertising program, and has also established a project management office to oversee the development of a new marketing career path for military personnel.
The report also identifies the actions taken by the Army to address the recommendations in the AAA report on contract oversight, including revising strategic goals, updating and documenting its assessment process, improving data systems, and integrating national and local marketing and advertising efforts. The report notes that the Army has taken steps to address the recommendations, but it is too early to determine the effectiveness of these efforts. The report concludes by stating that it will be important for the Army to continue to use commercial best practices for assessing the effectiveness of its marketing and advertising program and to regularly evaluate and improve its program to ensure its success. |

| Input length | Mesa-Extrapolation generate summary texts within 1000 tokens |
|---|---|
| 10k | The report discusses the challenges faced by the US Department of Veterans Affairs (VA) in providing disability compensation to veterans. The VA pays monthly disability benefits to veterans who have service-connected conditions, which are conditions that were caused or aggravated by military service. The VA has been facing challenges in determining when to reevaluate these conditions, and the report highlights several areas where improvements can be made. |
| | Firstly, the report notes that the VA does not have a clear understanding of the health outcomes of veterans with service-connected conditions. Despite the fact that the VA spends billions of dollars each year on disability compensation and health care for veterans, it does not have adequate data to assess the effectiveness of these efforts. The report recommends that the VA should develop a plan to collect and analyze data on the health outcomes of veterans with service-connected conditions. |
| | Secondly, the report highlights the issue of unwarranted reevaluations. Despite the fact that the VA has a rating schedule that provides guidelines for evaluating the severity of service-connected conditions, claims processors are not always following these guidelines. As a result, many veterans are undergoing unnecessary reevaluations, which can be time-consuming and costly. The report recommends that the VA should clarify the knowledge, skills, and abilities required for claims processors to determine when to reevaluate veterans' service-connected conditions. |
| | Thirdly, the report notes that the VA's procedures for reevaluating service-connected conditions are not always consistent. While the VA has a rating schedule that provides guidelines for evaluating the severity of service-connected conditions, claims processors are not always following these guidelines. As a result, many veterans are undergoing unnecessary reevaluations, which can be time-consuming and costly. The report recommends that the VA should develop and implement a plan to ensure that claims processors have the necessary knowledge, skills, and abilities to determine when to reevaluate veterans' service-connected conditions. |
| | Fourthly, the report highlights the issue of inconsistent application of rating criteria. Despite the fact that the VA has a rating schedule that provides guidelines for evaluating the severity of service-connected conditions, claims processors are not always applying these criteria consistently. As a result, many veterans are receiving inadequate compensation for their service-connected conditions. The report recommends that the VA should clarify the knowledge, skills, and abilities required for claims processors to apply the rating criteria consistently. |
| | Fifthly, the report notes that the VA's training program for claims processors is not adequate. While the VA provides training on disability compensation and health care for veterans, it does not provide enough training on the reevaluation process. As a result, many claims processors are not adequately equipped to determine when to reevaluate veterans' service-connected conditions. The report recommends that the VA should develop and implement a plan to provide additional training to claims processors on the reevaluation process. |
| | Finally, the report highlights the issue of limited data on the health outcomes of veterans with service-connected conditions. Despite the fact that the VA spends billions of dollars each year on disability compensation and health care for veterans, it does not have adequate data to assess the effectiveness of these efforts. The report recommends that the VA should develop a plan to collect and analyze data on the health outcomes of veterans with service-connected conditions. |
| | In summary, the report highlights several areas where the VA can improve its disability compensation program for veterans, including collecting and analyzing data on the health outcomes of veterans with service-connected conditions, clarifying the knowledge, skills, and abilities required for claims processors to determine when to reevaluate veterans' service-connected conditions, developing a plan to provide additional training to claims processors on the reevaluation process, and ensuring that claims processors are applying the rating criteria consistently. |

| Input length | Mesa-Extrapolation generate summary texts within 1000 tokens |
| --- | --- |
| 11k | The main content of the report is the analysis of the fiscal pressures faced by state and local governments in the United States. The report highlights several areas of concern, including:
1. Revenue growth: The report notes that state and local governments have seen a slowdown in revenue growth in recent years, which could lead to fiscal pressures.
2. Expenditure growth: The report also highlights the growing expenditures in areas such as healthcare, education, and infrastructure, which could put pressure on state and local government budgets.
3. Federal policy: The report notes that federal policies, such as the Tax Cuts and Jobs Act, could have a significant impact on state and local government fiscal pressures.
4. Natural disasters: The report notes that natural disasters, such as hurricanes and wildfires, can have a significant impact on state and local government budgets, particularly in terms of increased spending on emergency response and recovery efforts.
5. Debt: The report notes that state and local governments have increased their debt levels in recent years, which could lead to fiscal pressures in the future.
6. Pension obligations: The report highlights the growing pension obligations faced by state and local governments, which could put pressure on their budgets in the future.
7. Infrastructure: The report notes that state and local governments face significant infrastructure needs, particularly in areas such as transportation and water treatment, which could put pressure on their budgets.
8. Economic uncertainty: The report notes that economic uncertainty, such as the impact of the COVID-19 pandemic, can have a significant impact on state and local government revenues and expenditures.
The report also provides data on the trends in state and local government revenues and expenditures, including the growth of Medicaid spending and the impact of federal policy on state and local government fiscal pressures. It also highlights the challenges faced by state and local governments in terms of natural disasters, debt, pension obligations, and infrastructure needs. Finally, it notes the potential impact of economic uncertainty on state and local government revenues and expenditures. |

Table 5 shows that Mesa-Extrapolation adeptly and systematically summarizes key points one by one across various input lengths. It showcases logical coherence and demonstrates commendable summarization capabilities. It's worth noting that the initial symbols used for sub-items beneath each summary vary, including symbols such as "*", "first, second,...", or the number "1, 2, ...", etc. We attribute this variability to the inherent content of the input text. LLM follows the expression format of the text itself and generates corresponding output. Additionally, Mesa-Extrapolation accurately incorporates punctuation and exhibits an automatic stopping mechanism for proper stopping.

## C.9 Evaluation of Phi-3-mini-128k-instruct Model on Ruler Datasets

We further conducted experimental validation on the Ruler datasets Hsieh et al. (2024), focusing on the single-keys NIAH task. The needle-in-a-haystack (NIAH) test assesses the ability to retrieve a specific piece of information (the "needle") from long distractor texts (the "haystack").

In this experiment, we employed the microsoft/Phi-3-mini-128k-instruct model, which has practical value in real-world applications and is officially claimed to support extrapolation lengths of up to 128k tokens. To evaluate performance, we gradually increased the input length by 32k increments, comparing the original model's results with those achieved using our Mesa-Extrapolation method.

As shown in Figure 10, the original Phi-3-mini-128k-instruct model can only extrapolate up to its official limit of 128k tokens, beyond which it fails. In contrast, when utilizing the Mesa-extrapolation method, the model successfully extrapolates to at least 192k tokens. It's important to note that this 192k limit is due to our current hardware resource constraints—attempting to extend beyond this length results in an out-of-memory error.

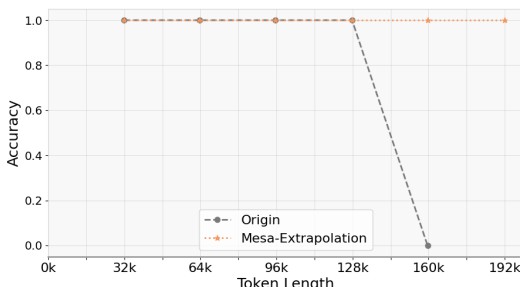

Figure 10: NIAH Task on Phi-3-mini-128k-instruct model using Origin and Mesa-Extrapolation.

# D Background

## D.1 Preliminary

In this section, we lay the groundwork and introduce the notation for below theorem analysis. Bold letters typically denote vectors or matrices, while normal letters represent scalars. We mainly follow Kazemnejad et al. (2023) to define these notations.

Let $f_\theta$ be a decoder-only Transformer model, where $\theta$ denotes the model parameters. $f_\theta$ processes the input sequence $x = [<bos>, x_1, \ldots, x_T]$ by applying a series of layers, where $<bos>$ represents the first token Tokenizer transformation.

Each layer $l$, consisting of self-attention heads and a feed-forward sub-layer, produces the hidden state $\boldsymbol{H}^l$ at layer $l$, after reading the previous hidden state $\boldsymbol{H}^{(l-1)}$.

Each head is parameterized by a query $\boldsymbol{W}_Q^m$, key $\boldsymbol{W}_K^m$, value $\boldsymbol{W}_V^m$, and output $\boldsymbol{W}_O^m$ matrices, where $m$ represents the attention head index, and $\boldsymbol{W}_Q^m$, $\boldsymbol{W}_K^m$, $\boldsymbol{W}_V^m \in \mathbb{R}^{h \times d}$ and $\boldsymbol{W}_O^m \in \mathbb{R}^{d \times h}$. $d$ is the model's hidden state size and $h$ is the attention dimension ($h = \frac{d}{\#heads}$). Note that we drop the attention head index $m$ where it is clear from the context.

The Transformer layer $\text{TLayer}^{(l)}(\boldsymbol{H}^{(l-1)}; \theta_l)$ consist of self-attention heads and a feed-forward sub-layer, and input the previous hidden state $H^{(l-1)}$, and generate the hidden state $H^{(l)}$ at layer $l$. $l$ is the layer index, and $\theta_l$ is the set of parameters of the $l$-th layer. Each hidden state $\boldsymbol{H}^{(l)} \in \mathbb{R}^{d \times (T+1)}$ is a matrix, and $\boldsymbol{h}_t^{(l)}$ denotes its hidden state at column $t$, i.e. at position $t$.

Each feed-forward sub-layer FF is parameterized by $\boldsymbol{W}_1$, $\boldsymbol{W}_2 \in \mathbb{R}^{d \times k.d}$ matrices, where $k$ denotes a multiplier of the hidden state size in this sub-layer, and is usually set to $4$ in common implementations of the Transformer.

The Transformer layer $\text{TLayer}^{(l)}$ processes each column of $\boldsymbol{H}^{(l-1)}$ independently and in parallel to generate the output. The computation of the $t$-th column of $\boldsymbol{H}^{(l)}$ is as follows:

$$\boldsymbol{h}_t^{(l)} = \text{FF}(\lambda(\boldsymbol{a}_t + \boldsymbol{h}_t^{(l-1)})) + \boldsymbol{a}_t + \boldsymbol{h}_t^{(l-1)}) \tag{4}$$

where $\lambda$ is layer normalization, and $\boldsymbol{a}_t \in \mathbb{R}^d$ is the output of the multi-head self-attention sub-layer at position $t$.

$\boldsymbol{a}_t$ is computed as:

$$\boldsymbol{a}_t = \sum_m \text{Attn}^{(m)}(\boldsymbol{h}_t^{(l-1)}, \boldsymbol{H}^{(l-1)}) \tag{5}$$

where $\text{Attn}^{(m)}$ denotes the $m$-th attention head. Let $\boldsymbol{o}_t \in \mathbb{R}^d$ denote the output of an attention head at position $t$. Then, $\boldsymbol{o}_t$ is computed as:

$$\boldsymbol{o}_t = \boldsymbol{W}_O \left( \sum_{i \leq t} \hat{\boldsymbol{\alpha}}_i \boldsymbol{v}_i \right) \tag{6}$$

where $\hat{\boldsymbol{\alpha}} = \text{softmax}(\boldsymbol{\alpha}) \in \mathbb{R}^{(t+1)}$, and $\boldsymbol{\alpha}$ is the attention weight vector such that:

$$\boldsymbol{\alpha} = [\langle \boldsymbol{q}_t, \boldsymbol{k}_0 \rangle, \ \langle \boldsymbol{q}_t, \boldsymbol{k}_1 \rangle, \ \ldots, \ \langle \boldsymbol{q}_t, \boldsymbol{k}_t \rangle]^T \tag{7}$$

where $\boldsymbol{q}_t = \boldsymbol{W}_Q \boldsymbol{h}_t^{(l-1)} \in \mathbb{R}^h$, $\boldsymbol{k}_i = \boldsymbol{W}_K \boldsymbol{h}_i^{(l-1)} \in \mathbb{R}^h$, and $\langle \cdot, \cdot \rangle$ denotes the dot product operation.

The feed-forward sub-layer $\text{FF}(\cdot) \in \mathbb{R}^d$ is a two-layer MLP as below:

$$\text{FF}(x) = \boldsymbol{W_2}\sigma(\boldsymbol{W}_1^{\text{T}}x) \tag{8}$$

where $\sigma$ is a non-linear activation function(usually ReLU or GeLU Hendrycks & Gimpel (2016)), and $\sigma(\cdot) \in \mathbb{R}^d$ is layer normalization Ba et al. (2016). It's worthy noted that the additive view of attention heads Elhage et al. (2021) in Eq.(5) is mathematically equivalent with the view of concatenate and multiple Vaswani et al. (2017). The additive view is easier to understand and analyze.

The hidden state $\boldsymbol{H}^{(0)}$ is initialized with a learned embedding of the input sequence $\boldsymbol{X}$, i.e. $\boldsymbol{H}^{(0)} = \boldsymbol{W}_E \boldsymbol{X}$, where $\boldsymbol{W}_E \in \mathbb{R}^{d \times V}$ is the embedding matrix and $\boldsymbol{X} \in \mathbb{R}^{V \times (T+1)}$ is the one-hot encoded input sequence. $V$ is the vocabulary size.

### D.2 NoPE & PE

**NoPE** We emphasize here that NoPE generally refers to these architectures that do not use position encoding components or remove the position encoding schemas, which is reflected in the attention dot product operation and is formally expressed as:

$$\langle \boldsymbol{q}_t, \boldsymbol{k}_i \rangle = \boldsymbol{q}_t^T \boldsymbol{k}_i \tag{9}$$

**PE** For PE, this generally refers to position encoding components, including Alibi, Rope, APE, etc. Their main difference from NoPE can be reflected in the attention dot product operation.

For ALiBi Press et al. (2021), it takes the form:

$$\langle \boldsymbol{q}_t, \boldsymbol{k}_i \rangle = \boldsymbol{q}_t^T \boldsymbol{k}_i - (t - i) \cdot C^{(m+1)} \tag{10}$$

where $m$ is head index and $C$ is a constant defined as:

$$C = 2^{-2^{-\log_2(\#\text{heads}+3)}} \tag{11}$$

For example, if the number of heads is 8, we have $\frac{1}{2}, \frac{1}{2^2}, \ldots, \frac{1}{2^8}$.

For RoPE Su et al. (2023), it's a relative PE that applies a rotation to the query and key representations based on their absolute positions before dot product attention. We formulate RoPE for model dimension $d = 2$, and its dot product as below:

$$\langle \boldsymbol{q}_t, \boldsymbol{k}_i \rangle = \boldsymbol{q}_t^T R^{(i-t)\theta} \boldsymbol{k}_i \tag{12}$$

where $R$ is a rotation matrix that rotates $(i - t)\theta$ radians:

$$R = \begin{bmatrix} \cos((i-t)\theta) & -\sin((i-t)\theta) \\ \sin((i-t)\theta) & \cos((i-t)\theta) \end{bmatrix} \tag{13}$$

For $d > 2$, RoPE applies the same approach on every two consecutive dimensions of $\boldsymbol{q}_t$ and $\boldsymbol{k}_i$, but with different $\theta$ angles.

For other PE methods, we recommend readers to consult the corresponding papers.

## E  Proofs

We provide proof about Theorems 3.1, 3.2, 3.3, and Corollary 4.1. Our proofs are inspired by Kazemnejad et al. (2023) Weiss et al. (2021) Lindner et al. (2023) and relies on the causal attention mask in the decoder-only Transformer and the SoftMax function.

Please refer to Appendix D for some necessary background knowledge.

> **Theorem E.1** (NoPE Extrapolation). *Let $x = [<bos>, x_1, \ldots, x_T]$ be an input sequence of length $T + 1$ to the model. Then, there exists $\boldsymbol{W}_Q$, $\boldsymbol{W}_K$, $\boldsymbol{W}_V$, $\boldsymbol{W}_O$, $\boldsymbol{W}_1$, and $\boldsymbol{W}_2$ matrices, such that when $T < M$, $o_T > \mathcal{H}$; and when $T > M$, $o_T < \mathcal{H}$.*

*Proof.* Our proof only specifies the weights of a single attention head in the first layer. In this parameterization, we only require the first three dimensions of the hidden states and regard the third dimension as the specific dimension. As for the remaining heads, they can be arbitrary as long as they don't override the first three dimensions. This doesn't pose any challenges, considering that Transformers used in practice usually have a very large model dimension $d$.

First, we construct the word embedding matrix $\boldsymbol{W}_E \in \mathbb{R}^{d \times V}$, where each column is the embedding of a token in the vocabulary. We construct $\boldsymbol{W}_E$ such that it always sets the first dimension of every embedding vector to be 1, and sets the second dimension to 0 except the token $<bos>$. We always assume $<bos>$ is the first token in the vocabulary, i.e. the first column. Then, we have:

$$\boldsymbol{W}_E = \begin{bmatrix} 1 & 1 & 1 & \ldots & 1 \\ 1 & 0 & 0 & \ldots & 0 \\ 0 & 0 & 0 & \ldots & 0 \\ e_{4,1} & e_{4,2} & e_{4,3} & \ldots & e_{4,V} \\ \vdots & \vdots & \vdots & \ldots & \vdots \\ e_{d,1} & e_{d,2} & e_{d,3} & \ldots & e_{d,V} \end{bmatrix}_{d \times V} \tag{14}$$

where $e_{d,i} \in \mathbb{R}$.

Then, we use the word embedding matrix $\boldsymbol{W}_E$ to compute the embedding $\boldsymbol{H}^{(0)}$:

$$\boldsymbol{H}^{(0)} = \boldsymbol{W}_E \boldsymbol{X} = \begin{bmatrix} 1 & 1 & 1 & \ldots & 1 \\ 1 & 0 & 0 & \ldots & 0 \\ 0 & 0 & 0 & \ldots & 0 \\ e_{4,1} & e_{4,2} & e_{4,3} & \ldots & e_{4,T+1} \\ \vdots & \vdots & \vdots & \ldots & \vdots \\ e_{d,1} & e_{d,2} & e_{d,3} & \ldots & e_{d,T+1} \end{bmatrix}_{d \times (T+1)} \tag{15}$$

Second, for head dimensions $h \geq 1$, we construct the weights $\boldsymbol{W}_K, \boldsymbol{W}_V, \boldsymbol{W}_O$ of the first attention head in the first layer. $\boldsymbol{W}_Q$ can be any arbitrary matrix. Specifically,

$$\boldsymbol{W}_K = \begin{bmatrix} 1 & 0 & \ldots & 0 \\ 1 & 0 & \ldots & 0 \\ \vdots & \vdots & \ddots & \vdots \\ 1 & 0 & \ldots & 0 \end{bmatrix}_{h \times d} \quad \boldsymbol{W}_V = \begin{bmatrix} 0 & M & \ldots & 0 \\ 1 - \mathcal{H} & 0 & \ldots & 0 \\ 0 & 0 & \ldots & 0 \\ \vdots & \vdots & \ddots & \vdots \\ 0 & 0 & \ldots & 0 \end{bmatrix}_{h \times d} \quad \boldsymbol{W}_O = \begin{bmatrix} 0 & 0 & 0 & \ldots & 0 \\ 0 & 0 & 0 & \ldots & 0 \\ 1 & -1 & 0 & \ldots & 0 \\ 0 & 0 & 0 & \ldots & 0 \\ \vdots & \vdots & \vdots & \ddots & \vdots \\ 0 & 0 & 0 & \ldots & 0 \end{bmatrix}_{d \times h} \tag{16}$$

$\boldsymbol{W}_K$ reads from the first dimension of the hidden state. Since all word embeddings have 1 in their first dimension, this parameterization will result all key vectors to be the same. Considering $\boldsymbol{k}_i = \boldsymbol{W}_K \boldsymbol{h}_i^{(0)}$, then:

$$\boldsymbol{k}_1 = \begin{bmatrix} 1 \\ 1 \\ \vdots \\ 1 \end{bmatrix}_{h \times 1}, \boldsymbol{k}_2 = \begin{bmatrix} 1 \\ 1 \\ \vdots \\ 1 \end{bmatrix}_{h \times 1}, \ldots, \boldsymbol{k}_{T+1} = \begin{bmatrix} 1 \\ 1 \\ \vdots \\ 1 \end{bmatrix}_{h \times 1} \tag{17}$$

We use $\boldsymbol{W}_Q$ to compute the query vector $\boldsymbol{q}_t$ by applying $\boldsymbol{q}_t = \boldsymbol{W}_Q \boldsymbol{h}_t^{(0)}$:

$$\boldsymbol{q}_t = [q_1, q_2, ..., q_h]^T \tag{18}$$

where $q_j \in \mathbb{R}$ can take any arbitrary value.

Next, we compute the attention weight vectors $\alpha$:

$$\alpha = [\langle \boldsymbol{q}_t, \boldsymbol{k}_1 \rangle, \langle \boldsymbol{q}_t, \boldsymbol{k}_2 \rangle, \cdots, \langle \boldsymbol{q}_t, \boldsymbol{k}_t \rangle]^T \tag{19}$$

$$= [\alpha^*, \alpha^*, ..., \alpha^*]^T \tag{20}$$

where $\alpha^* = q_1 + q_2 + \cdots + q_h$.

Then, we apply SoftMax to compute the attention weight score. Since all $\alpha^*$ are the same, we have:

$$\hat{\alpha} = \text{SoftMax}(\alpha) = [\frac{1}{t}, \frac{1}{t}, \ldots, \frac{1}{t}]^T \tag{21}$$

Now, we compute the value vectors by applying $\boldsymbol{v}_i = \boldsymbol{W}_V \boldsymbol{h}_i^{(0)}$:

$$\boldsymbol{v}_1 = \begin{bmatrix} M \\ 1 - \mathcal{H} \\ \vdots \\ 0 \end{bmatrix}_{h \times 1}, \boldsymbol{v}_2 = \begin{bmatrix} 0 \\ 1 - \mathcal{H} \\ \vdots \\ 0 \end{bmatrix}_{h \times 1}, \ldots, \boldsymbol{v}_t = \begin{bmatrix} 0 \\ 1 - \mathcal{H} \\ \vdots \\ 0 \end{bmatrix}_{h \times 1} \tag{22}$$

Then, we compute the attention value:

$$\sum_{i \leq t} \hat{\alpha}_i \boldsymbol{v}_i = \frac{1}{t} \sum_{i \leq t} \boldsymbol{v}_i = [\frac{M}{t}, 1 - \mathcal{H}, \ldots, 0]^T \tag{23}$$

Finally, we compute the output of the attention head by applying $W_O$:

$$\boldsymbol{o}_t = \boldsymbol{W}_O \left( \sum_{i \leq t} \hat{\alpha}_i \boldsymbol{v}_i \right) = [0, 0, \frac{M}{t} - 1 + \mathcal{H}, \ldots, 0]^T \tag{24}$$

$\square$

From Equ.24, it can be observed that when $t < M$, the hidden state value on the third dimension is greater than $\mathcal{H}$, indicating a successful extrapolation. Conversely, when $t > M$, the hidden state value on the third dimension is less than $\mathcal{H}$, indicating a failed extrapolation.

Next, we provide the proof for Theorem 3.2 as below:

> **Theorem E.2** (PE Extrapolation). *Let $x = [< bos >, x_1, \ldots, x_T]$ be an input sequence of length T+1 to the model. Consider an simple relative PE schema where dot product between query $\boldsymbol{q}_t$ and key $\boldsymbol{k}_i$ at positions t and i ($t \geq i$) can be expressed as: $\langle \boldsymbol{q}_t, \boldsymbol{k}_i \rangle := \boldsymbol{q}_t^T \boldsymbol{k}_i - (t-i)$. Then, there exists $\boldsymbol{W}_Q$, $\boldsymbol{W}_K$, $\boldsymbol{W}_V$, $\boldsymbol{W}_O$, $\boldsymbol{W}_1$, and $\boldsymbol{W}_2$ matrices, such that when $T < M$, $o_T > \mathcal{H}$; and when $T > M$, $o_T < \mathcal{H}$.*

*Proof.* We regard the third dimension as a specific dimension. And there is a threshold in this dimension in the second layer. Our proof construct the matrices on the first two layers. The first layer is used to extract position information. The second layer will generate a significant hidden state value in its third dimension. If the input length exceeds the max training length, the hidden state value will exceed the threshold.

**(1)Consider the first layer operation.**

First, following Theorem E.1, we construct the same word embedding matrix $\boldsymbol{W}_E$.

For head dimensions $h \geq 3$, we construct the weights $\boldsymbol{W}_K$ and $\boldsymbol{W}_V$ for the first attention head in the first layer. $\boldsymbol{W}_Q$ still can be any arbitrary matrix. Specifically,

$$\boldsymbol{W}_K = \begin{bmatrix} 0 & \cdots & 0 \\ \vdots & \ddots & \vdots \\ 0 & \cdots & 0 \end{bmatrix}_{h \times d}, \boldsymbol{W}_V = \begin{bmatrix} 0 & 1 & \cdots & 0 \\ 0 & 0 & \cdots & 0 \\ \vdots & \vdots & \ddots & \vdots \\ 0 & 0 & \cdots & 0 \end{bmatrix}_{h \times d} \tag{25}$$

Since the elements in $\boldsymbol{W}_K$ is all 0, by applying $\boldsymbol{k}_i = \boldsymbol{W}_K \boldsymbol{h}_i^{(0)}$, we have:

$$\boldsymbol{k}_1 = \begin{bmatrix} 0 \\ 0 \\ \vdots \\ 0 \end{bmatrix}_{h \times 1}, \boldsymbol{k}_2 = \begin{bmatrix} 0 \\ 0 \\ \vdots \\ 0 \end{bmatrix}_{h \times 1}, \ldots, \boldsymbol{k}_{T+1} = \begin{bmatrix} 0 \\ 0 \\ \vdots \\ 0 \end{bmatrix}_{h \times 1} \tag{26}$$

We use $\boldsymbol{W}_Q$ to compute the query vector $\boldsymbol{q}_t$ by applying $\boldsymbol{q}_t = \boldsymbol{W}_Q \boldsymbol{h}_t^{(0)}$:

$$\boldsymbol{q}_t = [q_1, q_2, ..., q_h]^T \tag{27}$$

where $q_j \in \mathbb{R}$ can take any arbitrary value.

Next, we compute the attention weight vectors $\alpha$. Consider $\boldsymbol{q}_t^T \boldsymbol{k}_i = 0$ and position information $t$ and $i$, we have:

$$\boldsymbol{\alpha} = [-(t-1), -(t-2), \ldots, 0]^T \tag{28}$$

Then, apply $\text{SoftMax}$ to $\boldsymbol{\alpha}$, we have:

$$\hat{\boldsymbol{\alpha}} = \text{SoftMax}(\boldsymbol{\alpha}) = [\frac{e^{-(t-1)}}{S}, \frac{e^{-(t-2)}}{S}, \ldots, \frac{e^0}{S}]^T \tag{29}$$

where define $S(t) = \sum_{j=0}^{t-1} e^{-j}$. We adopt its abbreviation $S$.

Now, we compute the value vectors by applying $\boldsymbol{v}_i = \boldsymbol{W}_V \boldsymbol{h}_i^{(0)}$:

$$\boldsymbol{v}_1 = \begin{bmatrix} 1 \\ 0 \\ \vdots \\ 0 \end{bmatrix}, \boldsymbol{v}_2 = \begin{bmatrix} 0 \\ 0 \\ \vdots \\ 0 \end{bmatrix}, \ldots, \boldsymbol{v}_t = \begin{bmatrix} 0 \\ 0 \\ \vdots \\ 0 \end{bmatrix} \tag{30}$$

Then, we compute the attention value:

$$\sum_{i \leq t} \hat{\alpha}_i \boldsymbol{v}_i = \frac{e^{-(t-1)}}{S} \boldsymbol{v}_1 = [\frac{e^{-(t-1)}}{S}, 0, \ldots, 0]^T \tag{31}$$

After that, we compute the output of the attention head by applying $\boldsymbol{W}_O$ (using the same construction as Equ.16 in Theorem E.1):

$$\boldsymbol{o}_t = \boldsymbol{W}_O \left( \sum_{i \leq t} \hat{\alpha}_i \boldsymbol{v}_i \right) = [0, 0, \frac{e^{-(t-1)}}{S}, \ldots, 0]^T \tag{32}$$

Then, $\frac{e^{-(t-1)}}{S}$ can be regarded as a monotonically decreasing function of $t$. Then through the MLP feedforward layer, we can always let the MLP layer recover the value of $t$.

Therefore, we can get that:

$$\boldsymbol{H}^{(l=1)} = \begin{bmatrix} 1 & 1 & 1 & \ldots & 1 \\ 1 & 0 & 0 & \ldots & 0 \\ 1 & 2 & 3 & \ldots & T+1 \\ e_{4,1} & e_{4,2} & e_{4,3} & \ldots & e_{4,T+1} \\ \vdots & \vdots & \vdots & \ldots & \vdots \\ e_{d,1} & e_{d,2} & e_{d,3} & \ldots & e_{d,T+1} \end{bmatrix}_{d \times (T+1)} \tag{33}$$

where position information is embedded in the third dimension of hidden state.

**(2)Consider the second layer operation.**

For head dimension $h \geq 3$, we construct the weights $\boldsymbol{W}_Q$ and $\boldsymbol{W}_K$ for the first attention head in the second layer. $\boldsymbol{W}_V$ follows the Theorem 3.1 setting. Specifically,

$$\boldsymbol{W}_Q = \begin{bmatrix} 0 & 0 & 0 & \ldots & 0 \\ 0 & 0 & 0 & \ldots & 0 \\ 0 & 0 & 1 & \ldots & 0 \\ q_{4,1} & q_{4,2} & q_{4,3} & \cdots & q_{4,d} \\ \vdots & \vdots & \vdots & \ddots & \vdots \\ q_{h,1} & q_{h,2} & q_{h,3} & \cdots & q_{h,d} \end{bmatrix}_{h \times d}, \boldsymbol{W}_K = \begin{bmatrix} 0 & 0 & 0 & \ldots & 0 \\ 0 & 0 & 0 & \ldots & 0 \\ 0 & 0 & -1 & \ldots & 0 \\ 1 & 0 & 0 & \ldots & 0 \\ \vdots & \vdots & \vdots & \ddots & \vdots \\ 1 & 0 & 0 & \ldots & 0 \end{bmatrix}_{h \times d} \tag{34}$$

Then, by applying $\boldsymbol{q}_i = \boldsymbol{W}_Q \boldsymbol{h}_i^{(0)}$, we have:

$$\boldsymbol{q}_1 = \begin{bmatrix} 0 \\ 0 \\ 1 \\ q_4 \\ \vdots \\ q_h \end{bmatrix}_{h \times 1}, \boldsymbol{q}_2 = \begin{bmatrix} 0 \\ 0 \\ 2 \\ q_4 \\ \vdots \\ q_h \end{bmatrix}_{h \times 1}, \ldots, \boldsymbol{q}_{T+1} = \begin{bmatrix} 0 \\ 0 \\ T+1 \\ q_4 \\ \vdots \\ q_h \end{bmatrix}_{h \times 1} \tag{35}$$

where $q_j$ can be arbitrary value for $j \geq 4$.

By applying $\boldsymbol{k}_i = \boldsymbol{W}_K \boldsymbol{h}_i^{(0)}$, we have:

$$\boldsymbol{k}_1 = \begin{bmatrix} 0 \\ 0 \\ -1 \\ 1 \\ \vdots \\ 1 \end{bmatrix}_{h \times 1}, \boldsymbol{k}_2 = \begin{bmatrix} 0 \\ 0 \\ -2 \\ 1 \\ \vdots \\ 1 \end{bmatrix}_{h \times 1}, \ldots, \boldsymbol{k}_{T+1} = \begin{bmatrix} 0 \\ 0 \\ -(T+1) \\ 1 \\ \vdots \\ 1 \end{bmatrix}_{h \times 1} \tag{36}$$

Then, we have:

$$\boldsymbol{q}_t^T \boldsymbol{k}_i = \sum_{4<=j<=h} q_j + (t - i) \tag{37}$$

Next, according to the PE definition, apply the position information $t$ and $i$ to the dot production, we have:

$$\langle \boldsymbol{q}_t, \boldsymbol{k}_i \rangle = \sum_{4<=j<=h} q_j$$

, which means $\langle \boldsymbol{q}_t, \boldsymbol{k}_i \rangle = \langle \boldsymbol{q}_t, \boldsymbol{k}_l \rangle$ for any $l \neq i$.

Then, we know that,

$$\boldsymbol{\alpha} = [\alpha^*, \alpha^*, \ldots, \alpha^*]^T \tag{38}$$

where $\alpha^* = \sum_{4 \leq j \leq t} q_j$.

Then, apply SoftMax to $\boldsymbol{\alpha}$, we have:

$$\hat{\boldsymbol{\alpha}} = \text{SoftMax}(\boldsymbol{\alpha}) = [\frac{1}{t}, \frac{1}{t}, \ldots, \frac{1}{t}]^T \tag{39}$$

Next, follow same process as Equ.(21) (22) (23) (24) in Theorem E.1, we can get same result in the second layer. It shows the limitation of extraordinary for position encoding. $\square$

Next, we provide the proof for Theorem 3.3 as below:

**Theorem E.3** (Weave PE Extrapolation). *Let $N$ be a positive constant. Consider a simple weave PE extrapolation schema: when $t - i < N$, $\mathcal{W}(t - i) = t - i$; and when $t - i \geq N$, $\mathcal{W}(t - i) = N$. Then, the attention dot product is fixed as below:*

$$\langle \boldsymbol{q}_t, \boldsymbol{k}_i \rangle := \begin{cases} \boldsymbol{q}_t^T \boldsymbol{k}_i - (t - i) & , \quad t - i < N \\ \boldsymbol{q}_t^T \boldsymbol{k}_i - N & , \quad t - i \geq N \end{cases}$$

*, where $N \ll M$. Then, applying $\boldsymbol{W}_Q, \boldsymbol{W}_K, \boldsymbol{W}_V, \boldsymbol{W}_O, \boldsymbol{W}_1,$ and $\boldsymbol{W}_2$ matrices from Theorem 3.2, we have when $T > M$, $o_T > \mathcal{H}$.*

*Proof.* Our proof is closely related to Theorem 3.2. We completely adopt the LLM setting by Theorem 3.2, and only modify the representation of the relative position. The purpose is to illustrate that effective extrapolation beyond the maximum window length can be achieved only by rearranging the relative position.

Following Theorem E.2, we can get $\boldsymbol{H}^{(0)}$.

Then, compute the attention weight vectors $\boldsymbol{\alpha}$. Consider $\boldsymbol{q}_t^T \boldsymbol{k}_i = 0$ and position information $t$ and $i$. By applying this **Extrapolation** schema, we have:

$$\boldsymbol{\alpha} = [-N, -N, \ldots, -1, 0]^T \tag{40}$$

$$= [-(t - (t - N)), -(t - (t - N)), \ldots, -1, 0]^T \tag{41}$$

Follow Eqs.28 and 29, and apply SoftMax to $\boldsymbol{\alpha}$, we have:

$$\hat{\boldsymbol{\alpha}} = \text{SoftMax}(\boldsymbol{\alpha}) = [\frac{e^{-(t-(t-N))}}{S^\#}, \frac{e^{-(t-(t-N))}}{S^\#}, \ldots, \frac{e^0}{S^\#}]^T \tag{42}$$

where $t > N$ and define $S^\#(t) = \sum_{j=0}^{N-1} e^{-j} + (t - N)e^{-N}$. We adopt the abbreviation $S^\#$ where there is no confusion.

Then, compute the value vector by applying $\boldsymbol{v}_i = \boldsymbol{W}_V \boldsymbol{h}_i^{(0)}$, still have:

$$\boldsymbol{v}_1 = \begin{bmatrix} 1 \\ 0 \\ \vdots \\ 0 \end{bmatrix}, \boldsymbol{v}_2 = \begin{bmatrix} 0 \\ 0 \\ \vdots \\ 0 \end{bmatrix}, \ldots, \boldsymbol{v}_t = \begin{bmatrix} 0 \\ 0 \\ \vdots \\ 0 \end{bmatrix} \tag{43}$$

Then, we compute the attention value:

$$\sum_{i \leq t} \hat{\alpha}_i \boldsymbol{v}_i = \frac{e^{-(t-(t-N))}}{S^\#} \boldsymbol{v}_1 = [\frac{e^{-(t-(t-N))}}{S^\#}, 0, \ldots, 0]^T \tag{44}$$

After that, we compute the output of the attention head by applying $\boldsymbol{W}_O$:

$$\boldsymbol{o}_t = \boldsymbol{W}_O \left( \sum_{i \leq t} \hat{\alpha}_i \boldsymbol{v}_i \right) = [0, 0, \frac{e^{-(t-(t-N))}}{S^\#}, \ldots, 0]^T \tag{45}$$

Notice that we set the MLP feed-forward process simulate a a piecewise function about $\frac{e^{-(t-1)}}{S(t)}$, which produce a Integer $t$ to recover the position.

Considering $t > N$, we know $\frac{e^{-(t-1)}}{S(t)} < \frac{e^{-(t-(t-N))}}{S^\#(t)} < \frac{e^{-(t-(t-N))}}{S(N+1)}$. And the $\frac{e^{-(t-(t-N))}}{S^\#(t)}$ function decreases extremely slowly as $t$ increases, almost equal to $\frac{e^{-(t-(t-N))}}{S(N+1)}$.

At the same time, the $\frac{e^{-(t-1)}}{S(t)}$ function decreases very quickly when $t$ is relatively small; As $t$ increases, it also decreases extremely slowly.

Combining the characteristics of these two functions, we can always choose a relatively small $N$ to keep it away from $M$ (that is $N \ll M$), so that the value recovered after $\frac{e^{-(t-(t-N))}}{S^\#(t)}$ passing through MLP process is around $N + 1$, because $\frac{e^{-(t-(t-N))}}{S(N+1)}$ can recover $N + 1$ and $\frac{e^{-(t-(t-N))}}{S^\#(t)}$ is almost equal to $\frac{e^{-(t-(t-N))}}{S(N+1)}$. For the sake of simplicity, considering the rounding characteristics of piecewise functions, we might as well set $\frac{e^{-(t-(t-N))}}{S^\#(t)}$ to recover $N + 1$.

Therefore, we can get the hidden state like that:

$$\boldsymbol{H}^{(l=1)} = \begin{bmatrix} 1 & 1 & \ldots & 1 & \ldots & 1 \\ 1 & 0 & \ldots & 0 & \ldots & 0 \\ 1 & 2 & \ldots & N+1 & \ldots & N+1 \\ e_{4,1} & e_{4,2} & \ldots & e_{4,3} & \ldots & e_{4,T+1} \\ \vdots & \vdots & \vdots & \vdots & \ddots & \vdots \\ e_{d,1} & e_{d,2} & \cdots & e_{d,3} & \ldots & e_{d,T+1} \end{bmatrix}_{d \times (T+1)} \tag{46}$$

**Continue the second layer operations.**

By applying $\boldsymbol{q}_i = \boldsymbol{W}_Q \boldsymbol{h}_i^{(0)}$, we have:

$$\boldsymbol{q}_1 = \begin{bmatrix} 0 \\ 0 \\ 1 \\ q_4 \\ \vdots \\ q_h \end{bmatrix}, \boldsymbol{q}_2 = \begin{bmatrix} 0 \\ 0 \\ 2 \\ q_4 \\ \vdots \\ q_h \end{bmatrix}, \ldots, \boldsymbol{q}_{(t>N)} = \begin{bmatrix} 0 \\ 0 \\ N+1 \\ q_4 \\ \vdots \\ q_h \end{bmatrix} \tag{47}$$

where $q_j$ can be arbitrary value for $j \geq 4$.

By applying $\boldsymbol{k}_i = \boldsymbol{W}_K \boldsymbol{h}_i^{(0)}$, we have:

$$\boldsymbol{k}_1 = \begin{bmatrix} 0 \\ 0 \\ -1 \\ 1 \\ \vdots \\ 1 \end{bmatrix}, \boldsymbol{k}_2 = \begin{bmatrix} 0 \\ 0 \\ -2 \\ 1 \\ \vdots \\ 1 \end{bmatrix}, \ldots, \boldsymbol{k}_{(i>N)} = \begin{bmatrix} 0 \\ 0 \\ -(N+1) \\ 1 \\ \vdots \\ 1 \end{bmatrix} \tag{48}$$

Then, consider $t > N$, and apply **Extrapolation**.

when $t - i < N$,
$$\langle \boldsymbol{q}_t, \boldsymbol{k}_i \rangle - f_{rel}(t - i) = \sum_{4 \leq j \leq h} q_j - (t - i) \tag{49}$$

when $t - i \geq N$ and $i > N$,
$$\langle \boldsymbol{q}_t, \boldsymbol{k}_i \rangle - f_{rel}(t - i) = \sum_{4 \leq j \leq h} q_j - N \tag{50}$$

when $t - i \geq N$ and $i <= N$,
$$\langle \boldsymbol{q}_t, \boldsymbol{k}_i \rangle - f_{rel}(t - i) = \sum_{4 \leq j \leq h} q_j + (N + 1 - i) - N \tag{51}$$

Let $\tau = \sum_{4 \leq j \leq h} q_j$. We can get the attention score:

$$\boldsymbol{\alpha} = [\underbrace{\tau - 0, \tau - 1, \ldots, \tau - (N - 1)}_{first-part}, \underbrace{\tau - N, \ldots, \tau - N}_{second-part}, \underbrace{\tau - (N - 1), \ldots, \tau - 1, \tau - 0}_{third-part}]^T \tag{52}$$

where $third-part$ represents Equ.49, $second-part$ represents Equ.50, and $first-part$ represents Equ.51.

Apply SoftMax to $\boldsymbol{\alpha}$, we can get that:
$$\hat{\boldsymbol{\alpha}} = \text{SoftMax}(\boldsymbol{\alpha}) = [\hat{\alpha_1}, \hat{\alpha_2}, \ldots, \hat{\alpha_t}]^T \tag{53}$$

Since the first element of $\boldsymbol{\alpha}$ is the maximum value, $\hat{\alpha_1}$ must be greater than $\frac{1}{t}$.

Next, similar process, we compute the value vectors by applying $\boldsymbol{v}_i = \boldsymbol{W}_V \boldsymbol{h}_i^{(0)}$:

$$\boldsymbol{v}_1 = \begin{bmatrix} M \\ 1 - \mathcal{H} \\ \vdots \\ 0 \end{bmatrix}, \boldsymbol{v}_2 = \begin{bmatrix} 0 \\ 1 - \mathcal{H} \\ \vdots \\ 0 \end{bmatrix}, \ldots, \boldsymbol{v}_t = \begin{bmatrix} 0 \\ 1 - \mathcal{H} \\ \vdots \\ 0 \end{bmatrix} \tag{54}$$

Then, we compute the attention value:

$$\sum_{i \leq t} \hat{\alpha_i} \boldsymbol{v}_i = [\hat{\alpha_1} \cdot M, 1 - \mathcal{H}, \ldots, 0]^T \tag{55}$$

Then, we compute the output of the attention head by applying $\boldsymbol{W}_O$:

$$\boldsymbol{o}_t = \boldsymbol{W}_O \left( \sum_{i \leq t} \hat{\alpha}_i \boldsymbol{v}_i \right) = [0, 0, \mathcal{H} + \hat{\alpha}_1 \cdot M - 1, \dots, 0]^T \tag{56}$$

When $t > M$, due to $\hat{\alpha}_1 > \frac{1}{t}$, it's possible to satisfy $\hat{\alpha}_1 \cdot M - 1 > 0$ condition, such that the hidden state value $o_t$ in the third dimension is greater than $\mathcal{H}$. $\qquad\square$

We conjecture that while the attention mechanism in RoPE Su et al. (2023) involves multiplying by the rotation matrix of the relative position, it's essential to acknowledge that neural networks possess the capability to model any function. This implies the potential to decompose the relative position information into an additive form, as exemplified in our extrapolation scheme detailed in Theorem 3.3.

Next, we provide the proof for Corollary 4.1 as below:

**Corollary E.4** (Mesa Extrapolation). *Let $N$ be a positive constant. Consider a simple Stair PE extrapolation schema, and the attention dot product is fixed as:*

$$\langle \boldsymbol{q}_t, \boldsymbol{k}_i \rangle := f_{\text{stairPE}}(\boldsymbol{q}_t, \boldsymbol{k}_i, t-i) = \begin{cases} \boldsymbol{q}_t^T \boldsymbol{k}_i - (t-i) & , & t-i < N \\ \boldsymbol{q}_t^T \boldsymbol{k}_i - I & , & t-i \geq N \end{cases}$$

*where $N \ll M$, $I = N + \left\lceil \frac{t-i-N}{E} \right\rceil$, and the extrapolated width $E$ is a constant. Then, Apply $\boldsymbol{W}_Q$, $\boldsymbol{W}_K$, $\boldsymbol{W}_V$, $\boldsymbol{W}_O$, $\boldsymbol{W}_1$, and $\boldsymbol{W}_2$ matrices from Theorem 3.2. Although $T > M$, it still $o_T > \mathcal{H}$.*

*Proof.* Due to the striking similarity between Stair PE and the positional arrangement scheme proposed in Theorem 3.3, the proof process largely mirrors Theorem 3.3. Following the proof structure of Theorem 3.3, in the first layer, according to Equ.45 and .46, we obtain the hidden state matrix as follows:

$$\boldsymbol{H}^{(l=1)} = \begin{bmatrix} 1 & 1 & \dots & 1 & \dots & 1 \\ 1 & 0 & \dots & 0 & \dots & 0 \\ 1 & 2 & \dots & I_N & \dots & I_{T+1} \\ e_{4,1} & e_{4,2} & \dots & e_{4,N} & \dots & e_{4,T+1} \\ \vdots & \vdots & \vdots & \vdots & \ddots & \vdots \\ e_{d,1} & e_{d,2} & \cdots & e_{d,N} & \dots & e_{d,T+1} \end{bmatrix}_{d \times (T+1)} \tag{57}$$

where $I$ is defined on Stair PE extrapolation schema.

In the proof of the second layer, we still get the similar result like Equ.52, as below:

$$\begin{aligned} \boldsymbol{\alpha} = [ & \underbrace{\tau - 0, \tau - 1, \dots, \tau - \mathcal{C}}_{\text{first-part}}, \\ & \underbrace{\tau - \mathcal{C}, \tau - \mathcal{C} - 1, \dots, \tau - N - 1}_{\text{second-part}}, \underbrace{\dots, \dots, \dots}_{\text{repeat-part}}, \\ & \underbrace{\tau - N - 1, \tau - \mathcal{C} - 1, \dots, \tau - \mathcal{C}}_{\text{third-part}}, \\ & \underbrace{\tau - \mathcal{C}, \dots, \tau - 1, \tau - 0}_{\text{fourth-part}}]^T \end{aligned} \tag{58}$$

where $\mathcal{C}$ is a constant, and each element within repeat-part is smaller than $\tau - 0$.

Apply $\text{SoftMax}$ to $\boldsymbol{\alpha}$, we can get that:

$$\hat{\boldsymbol{\alpha}} = \text{SoftMax}(\boldsymbol{\alpha}) = [\hat{\alpha}_1, \hat{\alpha}_2, \dots, \hat{\alpha}_t]^T \tag{59}$$

which also shows that the first element of $\boldsymbol{\alpha}$ is the maximum value and $\hat{\alpha}_1$ must be greater than $\frac{1}{t}$. Consequently, following Equ.56, we reach the same conclusion.

$\square$

# F   Probe Experiment Visualization

We hypothesize that when the input length surpasses the effective window length of the model, some dimensions' values in the exceeded positions will experience a jump as the position changes.

To investigate this jump phenomenon's correlation with extrapolation failure, we design the following experiment: we adopt a standardized input by repeating the word "hello" 8000 times, resulting in 8001 tokens (automatically fill in initial token) after tokenization. For a more accurate explanation, we take the LLaMA2-7B-Chat model and list the sequences converted by the tokenizer, as follows:

$$tokens = [1, 22172, \ldots, 22172]$$

where 1 denotes the initial token $< bos >$, and 22172 denotes the word "hello". It's noted that the initial token $< bos >$ is filled in automatically.

## F.1   Normal Case

We input this token sequence into the model and observe the hidden state. We focus on the hidden states produced by the first 11 layers, specifically selecting the position intervals from 4000 to 5000 and the last 1000 positions. We then concatenated the hidden states from these two intervals. This selection was deliberate, considering that the LLaMA2-7B-Chat model's training length is 4096, implying the effective input window is in proximity to this location.

Following this, we created a matrix graph, yielding the following results Fig.11:

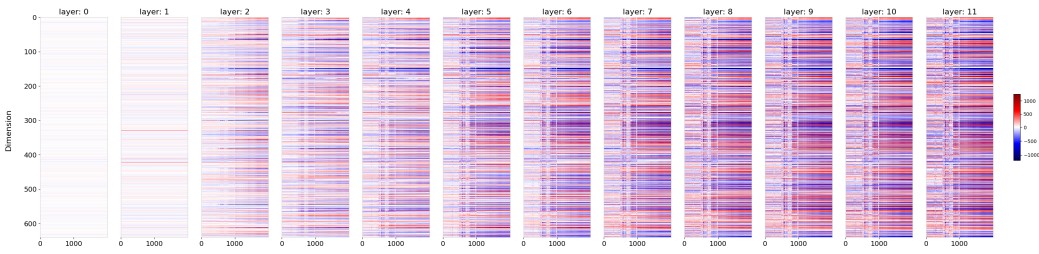

Figure 11: Origin with normal input token visualization on LLaMA2-7B-Chat model for position regions between (4000-5000, 7000-8000) for first 11 layers. X-axis represent positions, which correspond position regions between 4000-5000 and 7000-8000. Y-axis represent Dimentions from first 0-th to first 640-th. Some observation: noticeable color jumps in most dimensions as position changes.

In Fig.11, the X-axis denotes the position of the input token, ranging from 0 to 1000, representing tokens at positions 4000 to 5000. The scale of 1000-2000 represents tokens at positions 7001 to 8001. The Y-axis signifies the token dimension, ranging from small to large. The LLaMA2-7B-Chat model has a total of 4096 dimensions, and we display the first 640 dimensions in Y-axis. Red color means greater than 0, and blue color means less than 0. Fig.11 shows some noticeable jump from 0-th to 640-th dimensions, especially at the 1000 scale for the original model.

## F.2   Extrapolation Case

Given the effectiveness of the extrapolation method ReRoPE in extending to lengths of up to 8k, we apply it to the LLaMA2-7B-Chat model. Employing the same conditions as those in the **Normal Case** settings, the results are as follows:

In Fig.12, the obvious jumping phenomenon is successfully suppressed. It can be seen that each dimension at different positions still maintains consistent values. This illustrates that by suppressing sudden changes in the values of these dimensions, extrapolation can be made successful even outside the effective window length.

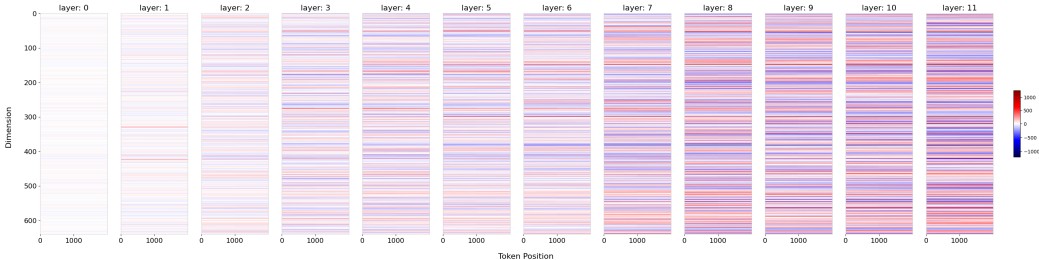

Figure 12: ReRoPE with normal input tokens visualization on LLaMA2-7B-Chat model for regions between (4000-5000, 7000-8000) for first 11 layers. X-axis represent positions, which correspond position regions between 4000-5000 and 7000-8000. Y-axis represent Dimentions from first 0-th to first 640-th. Observation: Each dimension stays consistently at each position

Overall, comparing **Normal Case** and **Extrapolation Case** again, it shows that using the extrapolation method can suppress mutations in dimensions, thereby making the extrapolation successful.

Additionally, for layers 0 and 1, the hidden state values of different positions in each dimension are nearly identical. According to our Theorem 3.2, layer 1 may extract position information, while subsequent layers determine distances beyond the effective window, generating a positive or negative jump signal. We speculate that this signal change may not be strictly positive or negative, but there is a threshold. When the signal exceeds the threshold, the extrapolation will fail; when the signal does not exceed the threshold, the extrapolation will be successful.

We also use the LLaMA-3B model and the Vicuna-13B model to compare and demonstrate the effects of using Origin and ReRoPE, as follows on below Fig.15 and .16.

### F.3 Validating extrapolation using observed thresholds

We continue our probe experiments using the "hello", and input sequences is up to 16000 tokens. These were validated on the LLaMA2-7B-Chat model. Given that the hidden state in LLaMA2 has a total of 4096 dimensions, we selected the first 20 dimensions to search observable thresholds. In the main text, we present the prediction results for the first and 6-th dimensions. The results for the 7-th and 9-th dimensions are shown in Figure 13.

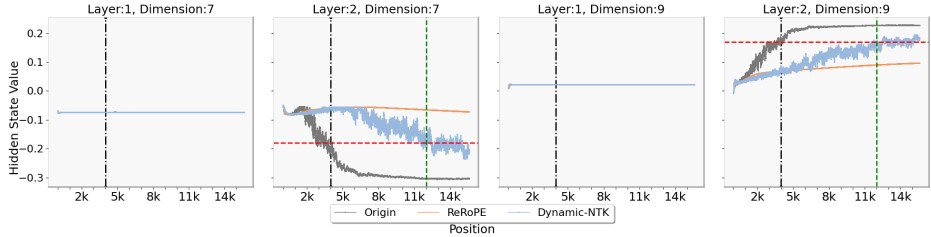

Figure 13: Thresholds for hidden states observed at specific dimensions on LLaMA2-7B-Chat, allowing for extrapolative judgments based on these thresholds. The red dashed line denotes the observed threshold, while the green dashed line indicates the position of extrapolation failure for the Dynamic-NTK based on the observed threshold. The black dashed line indicates the maximum training length of the model.

Figure 13 also shows the same results as Figure 2 in the main text. When the length of the input sequence is around 12k, the hidden state values of Dynamic-NTK in the 7-th and 9-th dimensions surpass the thresholds, implying extrapolation failure. Due to not outside the thresholds, ReRoPE shows successful extrapolation.

Our experiments also reveal that not every dimension exhibits a clear threshold. Considering that the Transformer model is a type of neural network, we hypothesize that these dimensions with threshold undergo significant signal changes due to the characteristics of activation functions. These changes are amplified through successive layers, eventually leading to model output failures. We believe

this internal mechanism explains why observable thresholds can predict the success or failure of extrapolation.

We also validate our findings using the Vicuna-13B model, and the results are shown in Figure 14:

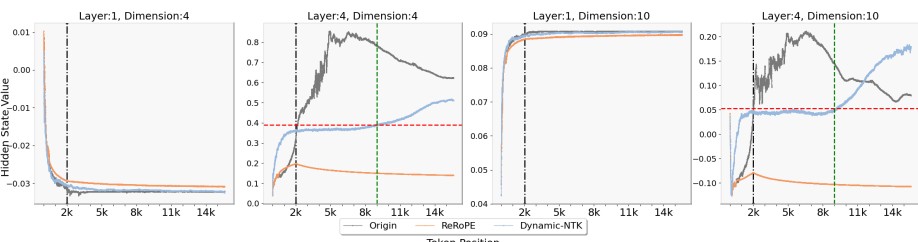

Figure 14: Thresholds for hidden states observed at specific dimensions on Vicuna-13B, allowing for extrapolative judgments based on these thresholds. The red dashed line denotes the observed threshold, while the green dashed line indicates the position of extrapolation failure for the Dynamic-NTK based on the observed threshold. The black dashed line indicates the maximum training length of the model.

Figure 14 shows the hidden state value on the specific 4-th and 10-th dimensions. Similar to the observations with LLaMA2-7B-Chat model, we find no significant differences for the hidden state values at the first layer. However, substantial differences are observed at the fourth layer. Based on these observed thresholds, we can predict that Dynamic-NTK will fail to extrapolate when the input length approaches around 9k. Conversely, ReRoPE can extrapolate further. These predictions align well with our subsequent experiments, further validating the feasibility of using observed thresholds to determine extrapolation success.

In contrast to LLaMA2-7B-Chat model, we observed threshold phenomena at the fourth layer in the Vicuna-13B model. We hypothesize that although our theoretical model predicts the threshold to occur at the second layer, considering the integration of positional information from preceding layers, this still aligns with our theoretical framework.

# G   Stair PE and Self-Extend

The formulation of Self-Extend is as follows:

Let $t$ denote the position of query, $i$ denote the position of key, $i$ denote the position of key, $W$ denote the neighbor size and $G$ denote group size.

According to its equation 3 in Self-Extend (Jin et al. (2024)), $P = P \mathbin{//} G$, and the shifted relative position $(W - W \mathbin{//} G)$, we can get the position of query as:

$$t \mathbin{//} G + W - W \mathbin{//} G$$

and the position of key as:

$$i \mathbin{//} G$$

By RoPE, their relative position between $t$ and $i$ (consider $t > i$), is remapped to:

$$t \mathbin{//} G + W - W \mathbin{//} G - i \mathbin{//} G = W + t \mathbin{//} G - i \mathbin{//} G - W \mathbin{//} G$$

It's worthy noting that only under the necessary and sufficient condition ( $t \bmod G >= i \bmod G$ ), we get that

$$t \mathbin{//} G - i \mathbin{//} G = (t - i) \mathbin{//} G$$

Next, when $W \bmod G == 0$, the equation of Self-Extend becomes:

$$W + (t - i) \mathbin{//} G - W \mathbin{//} G = W + (t - i - W) \mathbin{//} G$$

If we adjust this flooring operation to ceiling operation, and replace N with W and E with G, then Stair-PE is equivalent to Self-Extend.

In summary, under conditions $t \bmod G \geq i \bmod G$ and $W \bmod G == 0$, as well as changing the flooring operation to ceiling operartion in Self-Extend, these two formulas are equivalent. Except for this condition, they yield slightly different results.

Note because $W$ and $G$ are constants, the condition $W \bmod G == 0$ can be met easily. **But $i$ and $t$ change with positions, therefore there will always be positions where the condition $t \bmod G \geq i \bmod G$ is not satisfied**.

For example, $t = 10, i = 5, G = 2$, but $10 \mathbin{//} 2 - 5 \mathbin{//} 2 \neq (10 - 5) \mathbin{//} 2$.

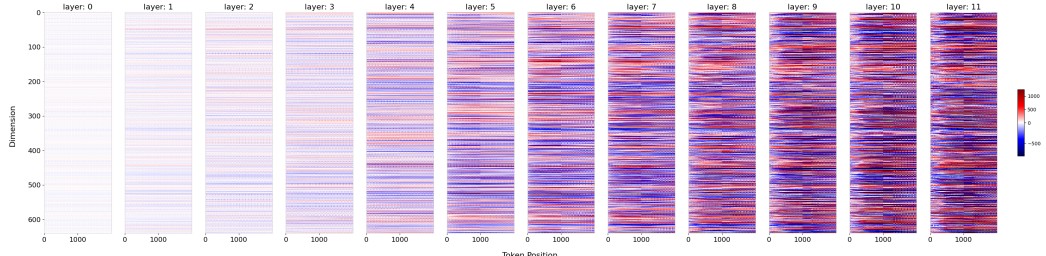

(a) Visualization of Hidden states with Origin Model for the first **11** layers and the first **640** dimensions. Observation: noticeable color jumps in most dimensions as position changes, especially at X-axis **1000**.

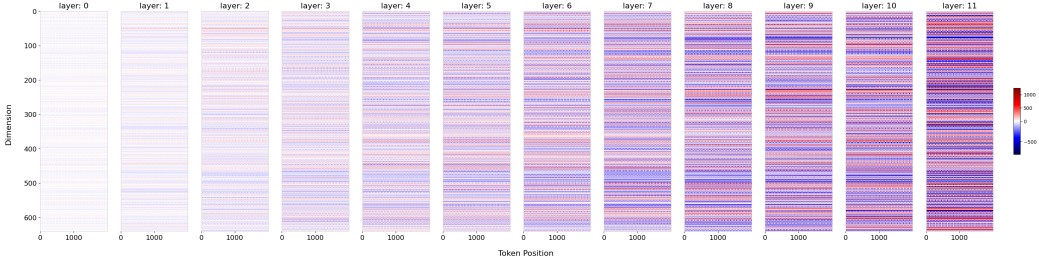

(b) Visualization of Hidden states with ReRoPE for the first **11** layers and the first **640** dimensions. Observation: Each dimension stays consistently at each position

Figure 15: Visualization of Origin and ReRoPE probe on LLaMA-3B model for position regions between (**2000-3000**, **4000-5000**)

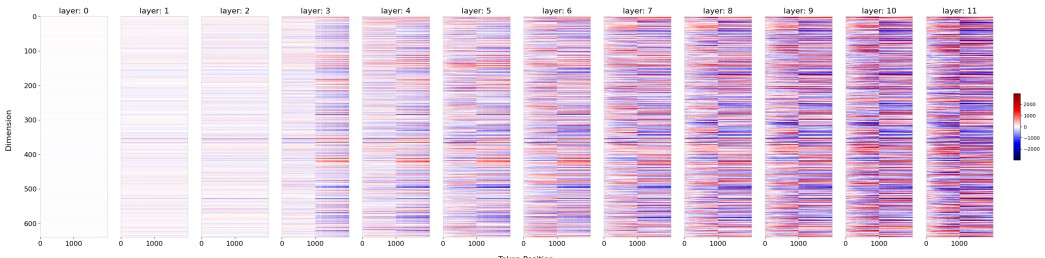

(a) Visualization of Hidden states with Origin Model for the first **11** layers and the first **640** dimensions. Observation: noticeable color jumps in most dimensions as position changes, especially at X-aixs **1000**.

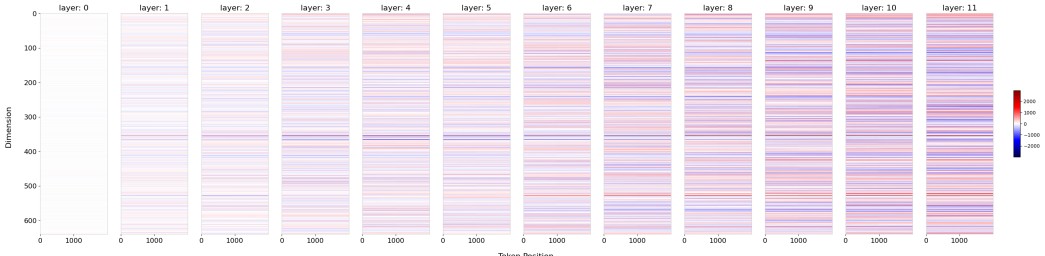

(b) Visualization of Hidden states with ReRoPE for the first **11** layers and the first **640** dimensions. Observation: Each dimension stays nearly same without color jump as position changes.

Figure 16: Visualization of Origin and ReRoPE probe on the Vicuna-13B model for position regions between (**2000-3000**, **4000-5000**)

