# OpenReview forum: "Mesa-Extrapolation: A Weave Position Encoding Method for Enhanced Extrapolation in LLMs"
_NeurIPS.cc/2024/Conference — NeurIPS 2024 poster_

### Official Review · Reviewer_vMAr · 2024-06-24

**Soundness:** 3
**Presentation:** 3
**Contribution:** 2
**Rating:** 5
**Confidence:** 5

**Summary:**

The paper conducts a theoretical analysis to help understand the No Position Encoding. Also, the paper proposes weave position encoding to achieve improved extrapolation performance without additional cost. Also, the paper introduces the weave PE method, Mesa-Extrapotion, which recalculates the position ID to reduce the gap between training and inference. Finally, the paper conducts experiments to prove the effectiveness of Mesa-Extropoation.

**Strengths:**

* The presentation of wave PE, Star PE, and Mesa-Extrapolation is clear. The author also provides the details of wave PE, Star PE and Mesa-Extrapolation to help understand the concepts
* The author conducts experiments to prove the effectiveness of the proposed Mesa-Extrapolation.
* The author also further analyzes the Latency & Memory Usage of the proposed Mesa-Extropoaltion.
* The paper discusses the limitations for further discussion.

**Weaknesses:**

**Major Concerns**: It seems that the proposed method Star PE is the same as Self-Extend LLM [1]. If possible, I sincerely hope that the author can address the following concerns:
* **Concern 1**: The Figure 1 Star PE implementation result does not match the Equation proposed in Section 4.1 Page 5. In Figure 5, when t-i is 5, the implementation result of Star PE is 4. However, according to the Equation proposed in Section 4.1 Page 5, the implementation result should be N+ $\lceil (t-i-N)/E \rceil$=4+$\lceil (5-4)/2 \rceil$=5. Hence, to match the implementation result of Figure 1, the Star PE calculation equation should be N+ $\lfloor (t-i-N)/2 \rfloor$.

* **Concern 2**: The Equation of Star PE is almost the same as Self-Extend LLM. When t-i is small than N, both Star PE and Self-Extend LLM employ normal relative distance. When t-i is larger or equal to N, we discuss it below.
   * The Equation of Star PE is N+ $\lfloor (t-i-N)/E \rfloor$ (as shown in Figure 1), while N is called the extrapolated position and E is called the extrapolated width, and t-i is the relative distance.
   * The Equation of Self-Extend LLM is $(t-i)//W + (W- W//G)$=$W+ (t-i)//G - W//G$, while W is called neighbor window size and G is called group size. Apparently, when W%G==0, the Equation of Self-Exntend LLM becomes $W+ (t-i)//G - W//G$=$W+ (t-i-W)//G$= $W+\lfloor (t-i-W)/G \rfloor$. Then, we change the notation W to N and the notation G to E, we have N+ $\lfloor (t-i-N)/E \rfloor$, which is the same as Star PE.
* **Concern 3**: If possible, could the author compare the performance between Mesa-Extropolation and Self-Extend LLM?
* **Concern 4**: when the output sequence length $L_{generate} \gg L_{input}$, will the time cost also becomes O($L_{generate}^2$)?

Based on the above concerns, the paper may need to rethink the major contribution. The proposed Mesa-Extrapolation seems to make sense and may benefit society, while the paper should clarify its original contribution.


Reference:

[1] Jin, H., Han, X., Yang, J., Jiang, Z., Liu, Z., Chang, C. Y., ... & Hu, X. (2024). Llm maybe longlm: Self-extend llm context window without tuning. arXiv preprint arXiv:2401.01325.

**Questions:**

Please see the above question and concerns in weakness.

**Limitations:**

Yes, the authors have addressed the limitations. We may further discuss and analyze the Mesa-Extropolation in other areas.

---

> ### Author Rebuttal · Authors · 2024-08-07
>
> Thanks for your work.
>
>
> ## major concern
>
> Thank you very much for your suggestions. The differences between our method and Self-Extend can be categorized into three aspects:
>
> Firstly, from a **methodological** perspective, our designed Stair PE is not only applicable to RoPE but also to other positional encoding methods, such as ALiBi. Self-Extend is only applied to RoPE. Additionally, Stair PE differs from the self-extend design approach, which involves implementing group attention and standard attention and then merging them through masking. This implies that it requires calculating the attention matrix twice, resulting in higher memory and time overhead. In contrast, our approach uses a chunk-based triangular attention matrix that significantly reduces computational and memory costs. We utilize Stair PE at the base of this chunk-based triangular attention matrix, and for practical applications, approximate methods can be employed to further save memory and reduce time costs.
>
> Secondly, **theoretically**, we provide a theoretical proof for the Weave PE approach, demonstrating that it can achieve extrapolation. In contrast, the self-extend approach only provides experimental evidence without a theoretical explanation.
>
> Finally, from an experimental standpoint, we further conducted comparisons of the two approaches and demonstrated that our method shows comparable advantages with lower memory consumption and inference latency (see attached PDF in Fig 5).
>
>
>
> ## concern 1
> We apologize for not checking it carefully. However, there is no need to modify the formula. We can simply change it to N=3.
>
>
> ## concern 2
> When we set the Stair PE formula as your derivation on # concern 1, your derivation is correct and aligns with Self-Extend LLM formulation. However, there is a crucial mismatch: **self-extend derives the positional encoding according this formulation just for its query vectors** (please refer to its pseudocode on algorithm 1 on its page 12).
>
> we briefly list below:
>
> ```
> g_pos = pos // g_size # the floor operation
> shift = w_size - w_size // g_size
> s_g_pos = g_pos + shift
> g_q = apply_pos_emcode (q , s_g_pos )
> g_k = apply_pos_emcode (k , g_pos )
> ```
>
> In its code, s_g_pos is correspond to your formula, but it’s not relative position encoding.
>
> Note that Self-Extend is only applied to RoPE positional encoding. Thus, with RoPE, the relative positional encoding is obtained by subtracting the key's positional encoding from the query's positional encoding. In contrast, our defined Stair PE **directly targets the relative positional encoding**. This is also why our Stair PE is more broadly applicable, not only to RoPE but also to other methods like ALiBi, whereas Self-Extend is only applicable to RoPE.
>
> According to these formulations, the relative positional encoding implemented by self-extend is different from that implemented by Stair PE. We have provided a clear comparison in the uploaded PDF (see Fig 6 ).
>
>
>
> ## concern 3
>
> The comparative results are provided in the uploaded PDF(see Fig 1, 2, 3, and 5).
> In both the NIAH and Passkey tasks (corresponding to Figure 3 in our paper), our method demonstrates comparable advantages. Additionally, our method performs well in terms of memory consumption and inference latency.
>
>
>
>
>
> ## concern 4
> The output of large models is divided into the prefill stage and the decoding stage. The time savings from mesa-extrapolation primarily occur in the prefill stage, which involves processing the N^2 matrix corresponding to the input sequence. During the subsequent token-by-token output process, the time complexity is O(L_generate + L_input ). This mainly concerns the last token and its computation with all preceding tokens, and this complexity is linear with respect to the sequence length.

---

> ### Comment · Reviewer_vMAr · 2024-08-08
> **Response to Author**
>
> Dear Authors,
>
> Thank you very much for your rebuttal. After reading the author's rebuttal, I still have questions about the contribution of Self-Extend and Star PE.
>
> * After the author's rebuttal, I confirm that the Star PE is equivalent to Self-Extend PE.
>    * First, their formulation is equivement, after adjusting floor and ceiling operations. **The core part is the same: increase 1 position number after $E$ steps and begin the operation at the $N^{th}$ step**.
>    * Then, the author claims that the Self-Extend PE cannot work on relative positional encoding. However, this is not correct
>       * The author may check Figure 2 in the Self-Extend LLM arxiv paper. This paper Figure 2 presents how they process the relative distances between queries and keys. Self-Extend can be directly applied to Alibi, as Alibi only needs to know the relative distance between queries and keys.
>
> * I appreciate the comparison between Mesa-Extropolation between Self-Extend. The result suggests the effectiveness of the Chunk-based Triangular Attention Matrix.
>
> To increase the score, I need the author to
> * Rethink the relation between Star PE and Self-Extend
> * Rewrite the part of Star PE.
> * Correctly claim the contribution of the paper, while I now have relatively high confidence to say that the Star PE and Self-Extend are equivalent. Therefore, the paper cannot claim that the Star PE is its contribution, while the theoretical analysis can be kept as part of contribution.
>
> Overall, I have to keep the current score and increase my confidence, as the paper contribution Star PE is equivalent to Self-Extend.

---

> > ### Author Response · Authors · 2024-08-09
> > **Part 1**
> >
> > Dear Reviewer  vMAr,
> >
> > Thank you very much for your valuable suggestions, which have been immensely helpful in improving the quality of our paper.
> > We apologize for not fully addressing your concerns, and provide further explanations in hope of clarifying a few details as mentioned.
> >
> > ## Rethink the relation between Star PE and Self-Extend
> > Based on careful examination of the respective resulting PEs, we conclude that the two methods yield similar outcomes. For example, under specific conditions (`t % G >= i % G`), your equations in concern 2 are valid and the resulting PEs are indeed equivalent. However, slight discrepancies between the resulting PEs emerge when such conditions are not met. We provide a detailed comparison at the end of this response.
> >
> > SelfExtend is a very recent work (published in July 2024), after our paper submission and current review process. And despite the striking similarities in the final PE results, our independent thought process was entirely different: We directly defined relative positions in Stair PE; while Self-Extend is defined by the original positions of query and key through group attention and neighbor attention.
> >
> > Furthermore, our theoretical contribution, as well as the novelty and effect of the Chunk-based Triangular Attention Matrix are valid and we believe add value to this field of study on LLM extrapolation.
> >
> > We will diligently articulate and clarify the relationship between Stair PE and Self-Extend in the revised version of the paper.
> >
> >
> > ## Rewrite the part of Stair PE
> > As advised, we will rewrite the Stair PE part (Section 4.1) in the final version, drafted as follows:
> >
> > “While our work was conducted independently, we note that Jin et al. have recently explored a similar idea through flooring the original positions and obtaining the relative position matrix with grouped self-attention. Although the two parallel thought processes are different, under certain conditions their formulations are equivalent (See appendix). Consequently, Self-Extend can be categorized as a Weave PE method. Our proposed Chunk-based Triangular Attention Matrix (detailed in Section 4.2) and its corresponding theoretical properties (Section 4.4) are also applicable to this parallel approach. ”
> >
> >
> >
> >
> > ## Our Contribution
> >
> > Following your suggestions, we will not specifically claim Stair PE as our main contribution in the paper. **Note in our introduction (line 45-60), we did not specifically claim Stair PE as our contribution either**. To further clarify, we will rewrite our claims as the following:
> >
> > 1.Theoretical Analysis: We provide a theoretical framework that explains extrapolation.
> >
> > 2.Introduction of Mesa-Extrapolation: which seamlessly integrates chunk-based triangular attention matrix and a Weave PE method (e.g. Stair PE).
> >
> > 3.Experimental Validation to showcase the effectiveness of Mesa-Extrapolation on a broad range of tasks.

---

> > > ### Author Response · Authors · 2024-08-09
> > > **Part 2**
> > >
> > > ## The detailed comparison of the formulas on Stair PE and Self-Extend
> > >
> > > The formulation of Self-Extend is as follows:
> > >
> > > let `t` denote the position of query, `i` denote the position of key, `W` denote the neighbor size and `G` denote group size.
> > >
> > > According to their paper’s equation (3) , `P = P // G`, and the shifted relative position (`W - W // G`), we can get the position of query as:
> > >
> > > `t // G + W - W // G                                                      (Equ.1)`
> > >
> > > and the position of key as:
> > >
> > > `i // G                                                                  (Equ.2)`
> > >
> > > By RoPE, their relative position between `t` and `i` (consider `t > i`), is remapped to:
> > >
> > > `t // G + W - W // G  -  i // G  =  W + t // G -  i // G - W // G                (Equ.3)`
> > >
> > > **Note this equation is slightly different from your equation in Concern 2**, which is `W + (t-i) // G - W // G`. **We would like to clarify that only under** `t % G >= i % G` **(necessary and sufficient conditions)**, we get `t // G - i // G` is equal to `(t-i) // G`.
> > >
> > >
> > > Next, when `W % G == 0`, the equation of self-extend becomes:
> > >
> > > `W + (t-i) // G - W //G = W + (t-i-W) // G `
> > >
> > > If we adjust this flooring operation to ceiling operation, and replace N with W and E with G, then Stair-PE is equivalent to Self-Extend.
> > >
> > > In summary, under conditions `t % G >= i % G` and `W % G == 0`, as well as changing the flooring operation to ceiling operartion in Self-Extend, these two formulas are equivalent.
> > > Except for this condition, they yield slightly different results.
> > >
> > > Note because `W` and `G` are constants, the condition `W % G == 0` can be met easily. But **`i` and `t` change with positions**, therefore there will always be positions where the condition **`t % G >= i % G`** is not satisfied.
> > >
> > > For example, `t = 10, i = 5, G = 2`, `10 // 2 - 5 // 2 != (10-5)//2`.
> > >
> > > This is why, as shown in the uploaded PDF(see fig 6), the relative positions differ between Self-Extend and Stair PE at certain positions (when `t % G < i % G` ).  Specifically, in Stair PE, the relative positions on the diagonal of the matrix always remain the same. In contrast, the diagonal values of the attention matrix in self-extend are not always the same.
> > >
> > > We will add discussions to the final version of our paper in appendix.
> > >
> > > Despite these subtle differences, we agree with your point that their core idea are the same.
> > >
> > > We are very grateful for your valuable suggestions and insightful observations, which have helped us think more deeply about their relationship. If you believe there are any issues with our analysis or have other concerns that haven't been addressed, please let us know, and we will make further improvements to enhance the quality of our work.
> > >
> > > Once again, thank you very much.

---

> > > > ### Comment · Reviewer_vMAr · 2024-08-10
> > > > **Response to Author**
> > > >
> > > > Dear Authors,
> > > >
> > > > Thank you very much for your reply. As the authors do not claim star pe as their contribution so that my concern of plagiarism is partially solved. Therefore, I increase the score from 4 to 5 and the confidence from 4 to 5.

---

> > > > > ### Author Response · Authors · 2024-08-10
> > > > >
> > > > > Dear Reviewer vMAr,
> > > > >
> > > > > Thank you very much for raising your score. We will express our gratitude for your valuable suggestions in the Acknowledgments Section of our paper.
> > > > >
> > > > > If you have additional concerns, please let us know.
> > > > >
> > > > > Once again, we thank you very sincerely.

---

### Official Review · Reviewer_bJoG · 2024-07-05

**Soundness:** 3
**Presentation:** 3
**Contribution:** 4
**Rating:** 8
**Confidence:** 3

**Summary:**

This paper studies the length extrapolation of LLMs.
1. It provides a theoretical analysis of why NoPE and PE fail to extrapolate beyond a certain length. Previous work has shown that this failure is related to the explosion of hidden states as positions increase. This paper demonstrates that both NoPE and PE suffer from this hidden state explosion, using a constructive approach to illustrate the existence of Transformer weights.
2. It proposes weave PE, a simple adaptation of PE that theoretically addresses the extrapolation issue. It also provides a simple implementation of weave PE, using a chunk-based triangular attention matrix. Then, it demonstrates that the proposed extrapolation scheme matches the performance of prior length extrapolation methods, such as Dynamic-NTK.

**Strengths:**

- Great theory explains the failure of NoPE and PE in length extrapolation.
- Proposes weave PE, derived from the theoretical analysis, which also works well in practice.
- Shows good empirical results in passkey retrieval, language modeling, and summarization.

**Weaknesses:**

1. Methodological comparison with $\Lambda$-Attention

The proposed Stair PE resembles the $\Lambda$-attention of LM-Infinite & Streaming-LLM, yet with differences in 1) the additional attention at the bottom, and 2) a different length extrapolation scheme, Meta-Extrapolation. In the experiments, Meta-Extrapolation significantly outperforms LM-Infinite & Streaming-LLM. Could the authors provide the intuition behind these empirical gains?

---
2. Empirical comparison with Dynamic-NTK

Dynamic-NTK outperforms Meta-Extrapolation on the summarization task for mid-lengths of 7-11k, while Meta-Extrapolation shows better performance on summarization for shorter lengths of 4-6k and better language modeling fluency for lengths greater than 11k. Could the authors provide the intuition behind these results?

---
3. Relation between input sequence length $T$ and effective length $M$

The theorems only show the existence of an effective length $M$, but do not provide intuition on the scale of $M$, such as the ratio over the input length $M / T$. Could the authors provide some intuition on this? If I understand correctly, $M$ is set from the construction of the Transformer weights, so can it be controlled to an arbitrarily large number?

---
Editorial comments

- The fonts of the figures and tables are too small. Please make them more readable.
- Some parts of the writing are mechanical. For example, lines 116-120 do not provide meaningful information. It would be great to discuss the implications of the theorems in natural language. For instance, both theorems state the failure of length extrapolation in NoPE and PE, rather than just "revealing the internal mechanism of extrapolation."

**Questions:**

See Weaknesses.

**Limitations:**

Well discussed. Extending this approach to fine-tuning would be an interesting next step.

---

> ### Author Rebuttal · Authors · 2024-08-07
>
> Thank you very much for recognizing our work and providing your suggestions.
>
> ## weakness 1
>
> For decoder-only architectures, the model's output is based on the next-token prediction. The last token in the input is crucial, as it generates the next token. For the last token to output the correct next token, it must attend to all previous tokens (principle of self-attention mechanism), especially those with critical information.
>
> The attention matrices used by Streaming-LLM and LM-Infinite both discard middle tokens, inevitably resulting in loss of information from these tokens. This problem becomes severe when the middle tokens carry important information.
>
> Our approach appends all tokens at the bottom, ensuring that information is not lost. For overly long inputs, we use Stair PE to reorganize relative position encoding, maintaining the integrity of the information.
>
>
>
> ## weakness 2
>
> We believe that the inherent capabilities of LLMs set the upper limit for these methods. As the input length increases, especially beyond the maximum training window, errors are bound to accumulate regardless of the method used, which will affect the model's performance.
>
> The NTK method scales the rotation angle of RoPE positional encoding. We hypothesize that within an 11k input length, the errors introduced by adjusting the rotation angle are acceptable for large models.
>
> For mesa-extrapolation, we reuse previous trained positions to minimize errors as much as possible. We speculate that with a fixed extrapolation width param (E = 50), the 7k-11k range may spread the model's attention more thinly compared to the 4k-6k range. For the summary task, the model needs to pay closer attention to the dispersed information within the context, which requires a finer extrapolation width to minimize errors. We hypothesize that optimizing the extrapolation width could alleviate or improve performance in the 7k-11k range.
>
>
> ## weakness 3
>
> Yes, M is set within the Transformer weights we constructed. This M is typically related to the maximum training window length of the model. For example, if the maximum window length for Llama is 2k, then M can be considered as 2k.
>
> Therefore, we think that M can't be controlled to an arbitrarily large number.
>
>
>
> ## Editorial comments
> Thank you for the modification suggestions. We made corresponding adjustments according to the comments.

---

> ### Author Response · Authors · 2024-08-12
>
> Dear Reviewer bJoG,
>
> We sincerely appreciate your positive feedback, and it is incredibly gratifying to have our work highly recognized by you.
>
> As the discussion progresses, if you have additional questions, please let us know, and we will be happy to provide further clarification.
>
> We will express our deep gratitude to you in the Acknowledgments section of our paper.
>
> Once again, thank you very much.

---

> > ### Comment · Reviewer_bJoG · 2024-08-12
> > **Response to the Rebuttal**
> >
> > Thank you for the rebuttal. I believe the paper is strong and will maintain my positive rating. However, I am not a strong expert in this area, so I will keep my original weak confidence.
> >
> > It would be helpful if the authors clarified answers to my questions in the revised paper. Specifically, the statement "M is typically related to the maximum training window length of the model" and thus "M can't be controlled to an arbitrarily large number" should be emphasized, as this significantly impacts the interpretation of the theorems.
> >
> > It's also interesting to observe how adding bottom attention enhances Streaming-LLM and LM-Infinite, especially when middle tokens contain critical information. Analyzing passkey retrieval based on its location could provide valuable insights, particularly if the improvement is most pronounced when the passkey is in the middle.

---

> > > ### Author Response · Authors · 2024-08-13
> > >
> > > Dear Reviewer bJoG,
> > >
> > > We greatly appreciate you for recognizing our work.
> > >
> > > ## the interpretation of Theorem
> > > Thank you for your valuable suggestions to improve the interpretation of the theorems of our paper. We will incorporate your suggestions into the final version of the paper.
> > >
> > > ## Enhancing Streaming-LLM / LM-Infinite
> > > We believe the question you proposed is very valuable.
> > > To address it, we suggest drawing our Mesa-Extrapolation design to add bottom-level attention for Streaming-LLM and LM-Infinite. Here’s how to add it:
> > >
> > > Input Tokens: |--------------------- Context Tokens ---------------------|---------- Last Tokens -------|
> > >
> > > #Operations : |<-----------Streaming-LLM / LM-Infinite----------->| <--- Bottom Attention-->|
> > >
> > >
> > > The diagram above illustrates how input tokens can be processed by dividing them into two parts: **context tokens** and **last tokens**.
> > >
> > > The context tokens are processed normally using Streaming-LLM or LM-Infinite, with the resulting key/value pairs being cached. When processing the last tokens, the last tokens’ key/value pairs can be appended with the cached key/value pairs(context tokens) to compute the attention using context forwarding (many operators, like Flash Attention, have already implemented this). During this context forwarding, if the keys are too long, the positions can be handled using the Weave PE methods (such as StairPE, ReRoPE, or Leaky-ReRoPE).
> > > We believe this approach can further enhance the capabilities of Streaming-LLM and LM-Infinite.
> > >
> > > We hope our responses addresses your questions. We will do our best to answer any further questions you may have before the final discussion deadline arrives.
> > >
> > > Once again, thank you very much.

---

### Official Review · Reviewer_PZAX · 2024-07-10

**Soundness:** 2
**Presentation:** 2
**Contribution:** 2
**Rating:** 4
**Confidence:** 4

**Summary:**

The paper proposes a positional embedding scheme to address the extrapolation issue: train on short sequences, evaluate on longer sequences. Authors propose a theoretical framing of the positional embeddings contribution to attention. They apply their analysis to NoPE (No Positional Embedding) and to standard PE, and RoPE. They propose the Mesa-Extrapolation idea where input tokens are organized so that attention is paid to nearby tokens and those at other key positions. Authors validate their findings with empirical evidence on several benchmarks and applications.

**Strengths:**

The paper is about a very relevant topic which has attracted a lot of attention lately. The paper proposes a simple approach to solve the problem which seems to be easy to adapt to different positional embedding models. Some of the numerical experiments are encouraging.

**Weaknesses:**

The theory part of the paper is hard to read and I am not sure about its usefulness. Result appear hand-wave-y and vaguely stated. For example the definition of the threshold H in the Assumption is surprising (see questions). Numerically, experiments on language modeling and Summary of Tasks do not seem to show the method's claims.

**Questions:**

1. Can authors explain the threshold definition: "When o > H, LLM extrapolates successfully. Once o < H, LLM extrapolation
fails." Is there a typo and inequalities are reversed?

2. In Fig 2, why dim 1 & 6 are of interest?

**Limitations:**

--

---

> ### Author Rebuttal · Authors · 2024-08-07
>
> Thanks for your work.
>
> ## Weakness 1 & question 1: “Theory is hard to read and unclear definition of the threshold H”
>
> **Definition of the extrapolation success or failure:**  When a large model continuously produces valid next-tokens for a given long input sequence, we define this as successful extrapolation. Conversely, if it outputs invalid next-tokens, we refer to this as failed extrapolation. Therefore, the key to extrapolation failure is the output of invalid next-tokens, which is caused by anomalies in the model's output. We speculate that these anomalies arise because certain parts of the model receive input values that exceed their acceptable range.
>
>
> **Definition of the threshold H:**  For convenience in our analysis, we consider the outputs of the preceding layer In a multi-layer neural network, which serves as the input for the next. To ensure normal subsequent outputs, the values from the preceding layer need to remain within a reasonable range. This implies that there must be a bound on these values. We **refer this observable boundary as the threshold H**. This threshold can be either an upper bound or a lower bound. In our work, for the sake of theoretical analysis, we chose the lower bound as our threshold.
>
>
> **Definition of hidden state values o**: the output of each layer in an LLM, which also serve as the input for the next layer.
>
>
> Note that in our assumption, we assume that for specific dimensions in specific layers, there exists an acceptable range of input values (i.e., hidden state values o). The lower bound of this acceptable range serves as our threshold H. By observing whether the hidden state value o in this dimension exceed the threshold H, we can predict whether the large model's extrapolation has failed.
>
>
> We apologize if it wasn't clear enough. We will add these refined definitions in the final version of the paper.
>
>
> ## weakness 2:
>
>
>
> Despite observing minor performance degradation in the 8k-11k token range for summarization tasks, our method consistently exhibits strong extrapolation capabilities across a diverse set of tasks, including passkey retrieval, language modeling, and summarization, as corroborated by Reviewers LfLf and bJoG.
>
> We also noted that the performance in the 4k-6k range is good. We speculate that with a fixed extrapolation width param (E = 50), the 7k-11k range may spread the model's attention more thinly compared to the 4k-6k range. For the summary task, the model needs to pay closer attention to the dispersed information within the context, which requires a finer extrapolation width to minimize errors. We hypothesize that optimizing the extrapolation width could improve performance in the 7k-11k range.
>
> ## Q1
>
> The inequalities in our Assumption are not reversed since we consider H as the lower bound.
>
> We will add clear definitions and explanations in the final version.
>
>
> ## Q2
>
> Dimensions 1 and 6, as well as dimensions 7 and 9 in the appendix (line 1246), were selected from the first 20 dimensions to showcase whether the extrapolation would fail using the observed threshold for NTK and Rerope. Note that different inputs activate different parts. Figure 2 used 'hello' as the probe word. Additionally, we demonstrated that dimensions 14 and 19 can be used to predict extrapolation failures using 'good' as the probe word (please refer to the uploaded PDF in Fig 7 and 8 for more detailed diagrams).

---

> ### Author Response · Authors · 2024-08-12
>
> Dear Reviewer PZAX,
>
> As Discussion Period progresses, we would like to know if our responses have addressed your concerns.
>
> We believe our responses address all of your main concerns; if they do, we would very appreciate it if you raised your score accordingly.
>
> If not, please let us know so that we can address any serious concerns that you might still have.
>
> Once again, thank you very much.

---

> > ### Comment · Reviewer_PZAX · 2024-08-12
> >
> > Thanks, I have read the rebuttal.
> >
> > I am still having trouble understanding Assumption 1's "If o > H extrapolation succeeds, if o < H, it fails": so in particular, take H = 100; we are talking about a model which for a sequence of length 10 < H = 100, extrapolation will fail, but with a long sequence of length (say) 100000000000000 > H, extrapolation succeeds. I have a hard time understanding what the assumption means here.
> >
> >
> > I am still feeling uncomfortable with the choice of dimensions 1 & 6 out of the top 20 dimensions. I do not see any reason why embedding vectors would be aligned with canonical basis vectors e_1 and e_{20}. I fear that the choices made here are very data specific and hard to reproduce in other setups.
> >
> > Another question: in the design of the chunk-based triangular attention, a strong assumption is made that tokens in the middle of the context are related to surrounding tokens and those at the beginning of the sequence. How did we validate this assumption?

---

> > > ### Author Response · Authors · 2024-08-13
> > >
> > > Dear Reviewer PZAX,
> > >
> > > Thanks for providing your further concerns.
> > >
> > > ## About Assumption 1
> > > We suspect there is a misunderstanding on the definition of o. In our assumption, "o" represents the hidden state value, not the sequence length of the input. Please refer to our first response for the definitions of "o" and "H".
> > > For example, consider an input sequence of length I with tokens denoted as below:
> > >
> > > input_tokens = $[x_0, x_1, x_2, …, x_I]$,
> > > where $x_i \in \mathbb{R}$
> > >
> > > After transformation by the Transformer matrix, we obtain the query vector, key vector, and value vector as below:
> > >
> > > Q = $[q_0, q_1, q_2…., q_I]$
> > >
> > > K = $[k0, k1, k2…., k_I]$
> > >
> > > V = $[v0, v1, v2, …v_I]$
> > >
> > > where $q_i, k_i, v_i \in \mathbb{R}^{d}$, e.g d =4096 for Llama2.
> > >
> > > Passing them to the Self-Attention part, we obtain that:
> > >
> > > P = $Q^T K \in \mathbb{R}^{I \times I}$
> > >
> > > O = $V \cdot Softmax(P)$ = $[O_0, O_1, O_2, ….., O_I]$
> > >
> > > where $O_i \in \mathbb{R}^{d}$, and let $O_i = [o_1, o_2, ..., o_{d}]^T$.
> > >
> > > Please note that $O_i$ corresponds to the hidden state values at position i. **We used $o$ to represent the hidden state value at a specific dimension**, that is,  **$o_j$ is the j-th element of $O_i$**.
> > >
> > > We assume that at a specific dimension (e.g. $o_j$ or dim j), there is a reasonable and acceptable range for this value. For simplicity, we have assumed a lower bound. If the value in this dimension falls below this lower bound, it could cause abnormal outputs in subsequent layers, ultimately leading to abnormal model behavior. This construction is intended to demonstrate that threshold-based criteria can be used to assess whether extrapolation is successful.
> > >
> > > ## About dimensions
> > >
> > > We suspect that there might be some confusion about dimensions in our context and canonical basis vectors.
> > >
> > > In our context, we use “dimension j” to describe the j-th element of the output vector $O_i$, as stated above . Therefore the concept of canonical basis vectors e_1 and e_{20} is not relevant in our context.
> > >
> > > Here we only chose specific examples of dimensions to illustrate the phenomenon of extrapolation failures. We observed that **this behavior is consistent** across different inputs and tasks (passkey, language modeling etc.). By observing the corresponding threshold, we can predict the extrapolation capability of the LLM, which is limited by its maximum pre-training window, regardless of the input.  Given the consistent results across various experimental conditions, we believe our findings have broad implications and are not data-specific.
> > >
> > >
> > >
> > > ## About the design of the chunk-based triangular attention
> > >
> > > First, our assumption that the beginning tokens and surrounding tokens are important is well-established and has been used in many of the previous works, such as Streaming-LLM and LM-Infinite. And the reasoning is below:
> > >
> > >  **Beginning of the sequence:**  1) people often place instructions, task descriptions, etc., at the start of a sentence, which is related to the way prompts are structured. 2) a start token is typically added by default, which is an important position holder. For instance, Llama2 automatically adds a <bos> token at the beginning. We’ve also explained this in detail in the appendix (line 1206-1209).
> > >
> > > **Surrounding tokens**: According to the principles of the attention mechanism, attention decreases as the distance increases, meaning the nearby tokens generally receive higher attention scores.
> > >
> > > Focusing on the initial tokens and their neighbors has been validated by other previous works (e.g,. LLM-Streaming and LM-Infinite). When designing the chunk-based triangular attention, we also experimented it and found this design to be the most effective.
> > >
> > > Note that Mesa-Extrapolation appends all previous tokens at the bottom of its attention matrix, allowing the last token to attend to all preceding tokens. In contrast, Streaming-LLM and LM-Infinite also discard middle tokens at the bottom of their attention matrix, inevitably resulting in loss of information from these tokens, leading to inferior performance.
> > >
> > > If you still have any questions regarding the explanation above, please let us know so that we can provide further clarification.
> > >
> > > Once again, thank you very much.

---

> > > > ### Comment · Reviewer_PZAX · 2024-08-13
> > > >
> > > > The assumption and notations are a lot more clear for me now. Observing / assuming that entry-wide hidden vector values matter is pretty interesting and surprising to me. Thanks for the explanation.

---

> > > > > ### Author Response · Authors · 2024-08-14
> > > > >
> > > > > Dear reviewer PZAX,
> > > > >
> > > > > your feedback is essential to us, if you think that our response addressed your concerns, we kindly ask you to consider raising your score. Please let us know if you have any questions or require further assistance.

---

> > > > > ### Author Response · Authors · 2024-08-14
> > > > > **Gently Reminder as the discussion deadline approaches**
> > > > >
> > > > > Dear Reviewer PZAX,
> > > > >
> > > > > We suspect that you might not be familiar with this type of observation and assumption. Similar observations also can be found in some related work, such as LM-Infinite (the attention logit bound in its Figure 1(a) ).
> > > > >
> > > > > We hope this is helpful.

---

### Official Review · Reviewer_hjGM · 2024-07-12

**Soundness:** 3
**Presentation:** 3
**Contribution:** 3
**Rating:** 6
**Confidence:** 4

**Summary:**

This paper introduces a new LLM length extrapolation method, called Mesa-extrapolation, which utilizes a chunk-based triangular attention matrix and applies stair PE. The proposed method is based on theoretical analysis. The paper conducts extensive experiments on passkey, PPL, summarization to demonstrate the effectiveness.

**Strengths:**

1. The paper provides a theoretical analysis to prove the effectiveness of meticulous weave position with PE for length extrapolation.
2. The proposed method is efficient and is proved to be effective through extensive experiments.

**Weaknesses:**

1. The passkey retrieval experiment is simple, good performance on the passkey is far from a real usable context window. Please consider to add evaluations on Ruler[1] and RepoQA[2]

2. The achieved context window is limited.



[1] https://arxiv.org/abs/2404.06654

[2] https://evalplus.github.io/repoqa.html

**Questions:**

1. Length extrapolation is a necessary technique, but the current extrapolation length is very limited. Considering that there are already many models that have undergone long context window extension, such as phi3-mini-128k, can your proposed method continue to perform length extrapolation on these long context LLMs? If so, it will significantly enhance the impact of your method

2. If I understand correctly, the proposed method is mainly for those with PE. Why is there a need to prove NoPE? Is NoPE your baseline?

3. The proposed Mesa-extrapolation is somehow similar as a variant of "sliding window attention + attention sinks". Could the author explain why mesa-extrapolation is theoretically superior compared to sliding window attention and attention sinks?

**Limitations:**

See the weakness and question sections.

---

> ### Author Rebuttal · Authors · 2024-08-07
>
> Thanks for your work.
>
> ## Weakness 1:
>
> We add evaluations on Ruler, and the results are available in the uploaded PDF (see Fig 1, 2 and 4). These results indicate that our method also performs well.
>
> ## Q1 & weakness 2:
> We perform experiments on long context window "microsoft/Phi-3-mini-128k-instruct" model using Ruler dataset. The results show that our method can further extrapolate to 192k, indicating the effectiveness of our method. We show it in the uploaded PDF (see Fig 4).
>
> Due to the inherent limitations of the phi3-mini-128k-instruct model itself when handling this task, even within a 128k window, the phi-3 model does not achieve 100% accuracy as shown in the right-side of Fig 4. Our method effectively extends its window based on the original model's capabilities, rather than improving its capability.
>
> ## Q2
> Although existing LLMs typically incorporate positional encoding (PE) components in their architecture, recent studies have shown that the NoPE (no positional encoding) method may better facilitate the extrapolation capabilities of LLMs. The choice of including or excluding the PE component in a LLM is critical, as it cannot be altered once the model is trained. We believe it is essential to theoretically model the extrapolation performance of both NoPE and PE. We discuss this part in our related work.
>
> Due to a lack of suitable candidates, NoPE was excluded from our baseline. Aside from the 1B-size pre-trained model (Kazemnejad, Amirhossein, et al. "The impact of positional encoding on length generalization in transformers." Advances in Neural Information Processing Systems 36 (2024).), we did not find any other suitable candidates. However, this pre-trained 1B model’s capabilities are very limited, making it difficult for evaluating basic NLP tasks.
>
> Additionally, by simply incorporating training-free weave PE into a PE-type LLM, we can achieve longer extrapolation. Through these efforts, we hope to demonstrate that choosing weave PE offers more advantages compared to NoPE.
>
>
>
>
> ## Q3
>
> Mesa-Extrapolation appends all previous tokens at the bottom of its attention matrix, allowing the last token to attend to all preceding tokens. When the input length significantly exceeds the model's maximum training window, it uses Stair PE to reuse positions and reduce errors further. The attention sliding window approach, on the other hand, discards tokens in the middle section. This method restricts the last token's attention to only the head and tail tokens, keeping the total attended token length within the maximum training window and directly using trained positions. This approach sacrifices the information from the middle tokens. In context-based tasks, the middle tokens often contain important information, and losing these tokens can prevent the model from providing accurate responses to user queries.

---

> ### Author Response · Authors · 2024-08-12
>
> Dear Reviewer hjGM,
>
> Thank you for your valuable suggestions and the time you’ve dedicated to our paper.
>
> As the discussion period draws to a close, we would like to know if our responses have addressed your concerns.
>
> Following your suggestion, we have conducted verification on phi3-mini-128k, which significantly enhance the impact of our approach.
>
> If our responses have resolved your concerns, we would greatly appreciate it if you could raise your score to a clear "accept".
>
> Once again, thank you very much.

---

> > ### Comment · Reviewer_hjGM · 2024-08-13
> > **Thank you for your rebuttal**
> >
> > Thank you for the rebuttal. The additional experiments addressed some of my concerns. However, I saw some performance drop between 32k-128k when extending phi3-128k. Therefore, I will increase my rating score to 6 accordingly.

---

> > > ### Author Response · Authors · 2024-08-13
> > >
> > > Dear Reviewer hjGM,
> > >
> > > Thank you very much for increasing your score. We will express our gratitude for your valuable suggestions in the Acknowledgments section of our paper.
> > >
> > > Additionally, regarding the performance decrease observed in the 32k-128k range, we would like to clarify further:
> > >
> > > First, our method can be applied only after 128k, not in the 32k-128k range, because it is inherently a free plug-in approach, and the effective input window of phi3-mini-128k itself can reach up to 128k.
> > > In practical scenarios, we would adopt our method only when the input exceeds 128k.
> > >
> > > Second, we have observed that the phi3-mini-128k model itself experienced a performance drop within the 32k-128k range, as shown on the right of Fig 4 (uploaded PDF), indicating its insufficiency in handling multi-key task within this range. Since our method is designed to assist the model in handling inputs beyond its effective 128k limit, not to enhance the model’s inherent capability, the performance issues beyond 128k are likely due to the inherent limitations of model for the multikey task, not to our method. In contrast, the Fig 4 left (uploaded PDF) shows that the phi3-mini-128k model itself successfully handles the singlekey task within the 128k range. Consequently, our method can extend its maximum effective input window beyond 128k, up to 192k.
> > >
> > > We hope our analysis effectively addresses your concerns and contributes to improved score. Please let us know if you have any further questions.
> > >
> > > Once again, thank you very much.

---

### Official Review · Reviewer_LfLf · 2024-07-15

**Soundness:** 3
**Presentation:** 2
**Contribution:** 3
**Rating:** 6
**Confidence:** 4

**Summary:**

The authors propose a weave position encoding method to enhance LLMs’ inference performance when the input context window exceeds the training context window. This method can be integrated into existing pretrained LLMs without additional finetuning. To support their findings, the authors conducted theoretical analyses on the failure reasons of various position encoding methods, including those without position encodings. They demonstrate that the significant shift in the hidden state’s value range, when input token positions exceed the maximum context length, is the cause of this phenomenon.

**Strengths:**

One of the strengths of the proposed method is that it can be integrated into existing pretrained LLMs without requiring any additional finetuning. This makes the method highly practical and easy to implement, saving both time and computational resources.

The method has demonstrated excellent performance in pass key retrieval tasks, showcasing its effectiveness in real-world applications. This indicates that the proposed approach not only works in theory but also delivers tangible improvements in practical scenarios.

The authors have conducted comprehensive theoretical analyses to understand the failure reasons of various position encoding methods, including those without position encodings. This thorough investigation provides a solid foundation for the proposed method and enhances its credibility

**Weaknesses:**

The proposed position encoding method, while promising, does not consistently improve performance across different tasks. This inconsistency suggests that the method may not be universally applicable or reliable in every context, potentially limiting its overall utility.

Additionally, the main narrative of the paper emphasizes the method’s ability to handle extrapolation beyond the training context window. However, given the observed variability in improvements, it would be more accurate to adjust the claims to better reflect the method’s performance, providing a more balanced and realistic presentation of the work.

**Questions:**

The caption for Figure 1 is not sufficiently informative.

Additionally, it is unclear how the failure of an LLM is measured in Section 3.4 and Figure 2.

The experiments visualizing hidden state values in Figure 2 would have been more effective if conducted on the same task and with the same setup as Figure 3. This alignment would allow for a clearer connection between the findings in Figures 2 and 3.

minor typos:

Theorem 3.2: an simple -> a simple

Line 161: defer to -> refer to

Line 242-243: a significant disruptions

**Limitations:**

The authors have included a limitations section; however, it reads more like a discussion of future work rather than addressing the actual limitations of the current study.

---

> ### Author Rebuttal · Authors · 2024-08-07
>
> Thanks for your work.
>
> ## weakness
> Thank you for your suggestions. Our method shows good extrapolation performance on accuracy-related tasks, but we observe slight variability in extrapolation performance within mid-length (8k-11k) in the summary task. We will adjust our claims accordingly in the final version.
>
> ## Q1
> Illustration of Mesa-Extrapolation. The left figure shows the Chunk-based triangular attention matrix (before SoftMax operation) of Mesa-Extrapolation while a sequence of length 13 is fed into an LLM. The right figure shows an example of PE and Stair PE. The Stair PE is used to weave the relative position equipped by Mesa-Extrapolation.
>
>
> ## Q2
> In sec 3.4, we designed a probe experiment by repeating a word (e.g. “hello”) N times as input to the model, where N is the input length (the details are provided in Appendix, line 1206-1209). The reason for designing the probe this way is to eliminate the influence of different tokens since both of the input token and its position affect the hidden state values.
>
> In Figure 2, we use a vertical black dashed line to indicate the position of maximum training length of the model. In this case, it is 4k for llama2-7b-chat model. The hidden state value at this position is designated as the observed threshold and marked with a horizontal red dashed line. When the hidden state value exceeds the red dashed line as the position changes, it signifies that the hidden state value has surpassed the threshold, **suggesting a failure in extrapolation** after that position. We will add these explanations to the final version.
>
> ## Q3
> We used a different experimental setup primarily for the following reasons:
> Figure 2 examines extrapolation failures, which primarily occur when input length surpasses the maximum limit, irrespective of the underlying task. Using the "hello" probe can clearly help us analyze the relationship between position and the corresponding thresholds. We then verify these predictions with other tasks, such as those depicted in Figure 3 or language modeling tasks shown in Figure 4. The results indicate that the threshold observed aligns with the results from the passkey task and language modeling task, demonstrating the effectiveness of the theory we developed.
>
> Nevertheless, we added additional results visualizing hidden state values using the same task and with the same setup as Figure 3 (see Fig 8 in attached PDF ).
>
> ## Q4
> Thanks. We fixed these typos.
>
>
> ## Limitations
> Due to limitations of resources, we have not yet validated our method at longer lengths. For instance, we have verified that phi3-mini-128k model can be extrapolated to at least 192k with our method. Beyond this length, memory crashes occur.

---

> > ### Comment · Reviewer_LfLf · 2024-08-12
> >
> > I appreciate the authors’ thorough rebuttal.
> >
> > The authors have effectively addressed all of my concerns and questions. Additionally, I have reviewed the other reviewers’ comments and the authors’ responses to those as well. Taking all these materials into account, I am increasing my final rating from 5 to 6. I believe the paper meets the criteria for acceptance as a poster presentation.

---

> > > ### Author Response · Authors · 2024-08-13
> > >
> > > Dear Reviewer LfLf,
> > >
> > > Thank you very much for raising your score, and we deeply appreciate the valuable suggestions you have provided. We will express our sincere gratitude to you in the Acknowledgments section of our paper.
> > >
> > > Once again, thank you very much.

---

> ### Author Response · Authors · 2024-08-12
>
> Dear Reviewer LfLf,
>
> As the discussion time goes by, we would like to know if our responses have addressed your concerns.
>
> If our responses have resolved your concerns, do you think our paper should now be a clear "accept"?
>
> We are very grateful for your valuable suggestions. If you have additional concerns, please let us know so that we can address them and further improve the quality of our paper.
>
> Once again, thank you very much.

---

### Author Rebuttal · Authors · 2024-08-07

Dear reviewers,
  Thank you very much for your review.
  We have provided additional experimental supplements in the uploaded PDF.
  Please check it out.

---

### Decision · Program_Chairs · 2024-09-25

**Decision:**

Accept (poster)

**Comment:**

The paper investigates position encoding techniques aimed at addressing the challenge of length extrapolation in Large Language Models (LLMs). The authors present a theoretical analysis of various extrapolation strategies, identifying the limitations of existing methods such as No Position Encoding (NoPE) and simple relative Position Encoding (PE) in extrapolation scenarios. To address these limitations, they propose a novel approach, the weave PE, which is designed to improve extrapolation performance. The theoretical contributions are well-supported by extensive empirical evaluations, offering significant insights and practical methods for improving LLMs.

During the review process, it was noted that the method referred to as "Stair PE" in Section 4.1 is analogous to a prior method known as Self-Extend PE (https://arxiv.org/abs/2401.01325). This issue was thoroughly discussed during the author-reviewer and reviewer-AC interactions. Despite this similarity, the consensus among the reviewers and the AC was that the paper provides substantial value to the research community, even without considering Section 4.1 as a novel approach. Therefore, the paper has been accepted for publication. The authors have been advised to explicitly acknowledge the connection between Stair PE and Self-Extend PE in the final manuscript to ensure clarity and transparency for the readers.

Additionally, the authors are requested to make the following edits: (1) Correct the usage of punctuation. For example, "i.e." in Line 37 should be "i.e.," the period should be placed within quotation marks in Line 52, and each equation should end with appropriate punctuation. (2) Increase the font sizes in some figures, as they are currently too small.